# Learning DAGs and Root Causes from Time-Series Data

## Abstract

We introduce DAG-TFRC, a novel method for learning directed acyclic graphs (DAGs) from time series with few root causes. By this, we mean that the data are generated by a small number of events at certain, unknown nodes and time points under a structural vector autoregression model. For such data, we (i) learn the DAGs representing both the instantaneous and time-lagged dependencies between nodes, and (ii) discover the location and time of the root causes. For synthetic data with few root causes, DAG-TFRC shows superior performance in accuracy and runtime over prior work, scaling up to thousands of nodes. Experiments on simulated and real-world financial data demonstrate the viability of our sparse root cause assumption. On S&P 500 data, DAG-TFRC successfully clusters stocks by sectors and discovers major stock movements as root causes.

## 1 Introduction

Many applications produce time-series data: multi-dimensional data measured in regular time steps. Examples include temperature measurements at different sites in meteorology (Yang et al., 2022), stock prices in finance (Kleinberg, 2013; Jiang &Shimizu, 2023), and brain data in medicine (Smith et al., 2011). A key problem in analyzing time-series data is causal structure discovery, which aims to understand the generation mechanism of such data between nodes and across time (Assaad et al., 2022b; Runge et al., 2023; Gong et al., 2023; Hasan et al., 2023). On common structural model associates time-series data with directed acyclic graphs (DAGs) that encode how the data in one time step is obtained from prior ones. Our work specifically focuses on learning these DAGs from time-series data (Sun et al., 2023; Gao et al., 2022; Pamfil et al., 2020). This approach simplifies the broader problem of causal discovery by abstracting away the need for true causal relationships, which often require techniques like interventions. Despite this simplification, DAG learning from time series still poses a challenge due to the complexity of temporal dependencies and the high dimensionality of data.

**Structural vector autoregression.** Prior work on structure discovery from time series parametrizes the data generation as structural vector autoregression (SVAR) (Lütkepohl, 2005; Pamfil et al., 2020), which was first introduced by Sims (1980) and is commonly used in econometrics (Kilian, 2013). SVAR describes linear dependencies between nodes that are either instantaneous, if the nodes are at the same time point, or lagged if there is a time delay between the nodes. The SVAR model implies stationarity, i.e., that these dependencies are the same for every time point. Together they form a DAG, called the window graph, which uniquely determines the parameters of the SVAR. Learning a window graph with time lags is a challenging task and requires more parameters than learning DAGs from static data, which is already an NP-hard problem (Chickering et al., 2004). Various methods to learn the (weighted) window graph from time-series data have been proposed, e.g., by Hyvärinen et al. (2010); Nauta et al. (2019); Pamfil et al. (2020). However, some methods, including Granger causality methods (Bussmann et al., 2021; Marcinkevičs &Vogt, 2020) ignore the time lags or do not consider instantaneous dependencies (Entner &Hoyer, 2010; Khanna &Tan, 2019). In addition, methods are generally inefficient to compute graphs with thousands of nodes (Cheng et al., 2024). Moreover, there is a limited interpretation of the input variables of the SVAR, which we will refer to as root causes. Here, we aim to make progress on those challenges with a novel efficient method based on the assumption of few root causes.

**Root causes.** In this paper we are concerned with learning the window graph under the assumption that the time series are generated from an SVAR with few root causes. Intuitively this means that there exist (approximately) few significant events on nodes and in time that propagate through the graph and time, as stipulated by the SVAR to produce the time-series data. This assumption differs from prior work in which the input is typically assumed as zero-mean random i.i.d. noise (Pamfil et al., 2020; Gao et al., 2022; Tank et al., 2021). Our motivation comes from the few root causes assumption for static data proposed by Misiakos et al. (2023b), successfully applied to gene activation data (Misiakos et al., 2023a). Our work expands the applicability of this assumption to the case of time series and, in addition, interprets the root causes in an experiment on real-world financial data. Note, that the root cause terminology we use is different but related to root cause analysis in (Ikram et al., 2022) as we explain in the related work later.

**Contributions.** In this paper we learn the weighted window graph associated with an SVAR for time-series data under the assumption of few root causes and, in addition, recover their values, locations and time points. Specifically:

- We formulate the few root causes assumption for time-series data generated from an SVAR. We show that the ground truth window graph is identifiable and is the global minimizer of the number of non-zero root causes for absent noise.

- We present DAG-TFRC, which learns the window graph from time-series data assuming an SVAR with few root causes. In synthetic experiments with few root causes, we show that DAG-TFRC can learn window graphs with up to several thousands of nodes and shows superior performance over various prior state-of-the-art methods.

- We show the viability of the few root cause assumption on simulated and real-world finance data. On simulated data (where the ground truth is known) the few root cause assumption achieves the best results. On real-world stock market data from the S&P 500 index, the few root cause assumption allows us to reasonably cluster the stocks within sectors and identify root causes that reflect significant changes in the stock prices.

## 2 SVAR AND ROOT CAUSES

We introduce notation, the needed background on SVARs, and the concept of root causes.

**Time-series data.** A multi-dimensional data vector $x_t$, measured at time point $t \in 0, 1, \ldots, T-1 = [T]$, is written as $x_t = (x_{t,1}, x_{t,2}, \ldots, x_{t,d}) \in \mathbb{R}^{1 \times d}$. A time series consists of a sequence of such data vectors $x_0, \ldots, x_{T-1}$ recorded at consecutive time points. We assume these vectors stacked as rows in a matrix, representing the entire time series, denoted as $X \in \mathbb{R}^{T \times d}$. When multiple realizations of $X$ are available, they are collected as slices of a tensor $\mathbf{X} \in \mathbb{R}^{N \times T \times d}$. These are typically obtained by dividing a long time series into smaller segments of length $T$.

**Example: stock market.** We consider an example of time-series data from the stock market. We collect daily stock values $x_t$ for a particular stock index (e.g., S&P 500) for, say, 20 years. A time series for one year is denoted with the matrix $X$ and 20 years yield the data tensor $\mathbf{X}$.

**Model demonstration.** We impose a graph model assumption on the generation of time-series data and demonstrate it first with a simple example. Assume that the vector $x_t$ at time $t$ is generated from the data $x_{t-1}$ of the previous time step according to the equation:

$$x_t = x_{t-1}B + c_t, \tag{1}$$

where $c_t$ are input variables, which we call *root causes* following Misiakos et al. (2023b), but have been also referred to as structural shocks (Kilian, 2013). The $(i, j)$ entry of the matrix $B \in \mathbb{R}^{d \times d}$ represents the influence of $x_{t-1,i}$ on $x_{t,j}$ and corresponds to the adjacency matrix of a directed graph $\mathcal{G} = (\mathbb{V}, B)$ where $\mathbb{V}$ is a set of nodes enumerated as $\mathbb{V} = \{1, 2, ..., d\}$. The goal is to learn $B$. The model (1) is stationary, since $B$ is the same for all $t$, and has time lag one, since $x_t$ is explained by the previous time step (and the new inputs at time $t$).

**Example.** In the stock market example, the stocks $1, 2, \ldots, d$ in the S&P market index would represent the nodes of a graph and $B$ would encode the influences between these stocks. The model then implies that the value $x_{t,i}$ of a stock $i$ on day $t$ is determined by the stock values $x_{t-1}$ from day $t-1$, combined with a root cause $c_{t,i}$ representing an event occurring on day $t$.

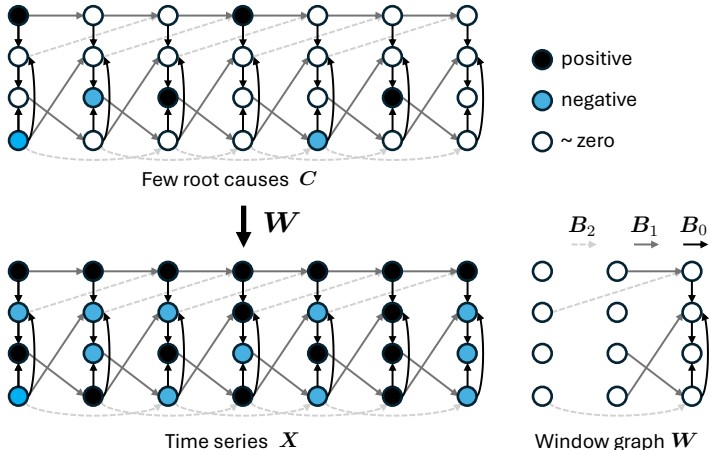

**Figure 1:** Generating time series $X$ with 7 time steps from an SVAR with few root causes $C$. Root causes (top) corresponding to nodes can be positive, negative or zero, with only a few being non-zero. There can be at most $4 \times 7 = 28$ root causes, and in our example only a few 7 are significant. The window graph $W$ consisting of $B_0, B_1, B_2$ produces the observed dense time series $X$ (bottom) with (4).

**Structural vector autoregression.** We expand (1) to the general form of an SVAR (Lütkepohl, 2005; Pamfil et al., 2020) with time lag $k$. Namely, we assume there exist adjacency matrices $B_0, B_1, ..., B_k \in \mathbb{R}^{d \times d}$ and $c_t \in \mathbb{R}^{1 \times d}, t \in [T]$, such that for all $t \in [T]$[1]:

$$x_t = x_t B_0 + x_{t-1} B_1 + ... + x_{t-k} B_k + c_t. \tag{2}$$

The $(i, j)$ entry of $B_\tau$ represents the influence of $i$ to $j$ after $\tau$ time-steps (i.e., a lag of $\tau$) and $c_t$ are the root causes. $B_0$ represents the *instantaneous* dependencies, while the $B_1, ..., B_k$ represent the *lagged* dependencies. The SVAR is *stationary*, since the $B_\tau$ do not depend on $t$.

As in prior work (Pamfil et al., 2020) we assume that $B_0$ corresponds to a DAG, which ensures that the recurrence (2) is solvable with respect to $x_t$.

The instantaneous $B_0$ and lagged dependencies $B_1, ..., B_k$ are collected as block-rows in a matrix $W \in \mathbb{R}^{d(k+1) \times d}$ which forms the so-called window graph[2][3] depicted with an example in Fig. 1. Note that the window graph is a DAG since the edges go only forward in time. The problem we aim to solve is to infer the window graph $W$ from time-series data under the assumption that there are few root causes. To achieve this, our approach imposes a sparsity assumption on the root cause terms.

**Example.** In the previous stock market example, $B_0$ would model instantaneous influences within the same day, and the other $B_\tau$ influences across days. In this case, one would expect most influences to be captured by $B_0$ since stock markets usually react close to instantaneously.

**Few root causes.** We denote with $x_{t,\text{past}} = (x_t, x_{t-1}, ..., x_{t-k})$, $t \in [T]$ the data at previous time steps of $x_t$ with lag up to a chosen fixed $k$. Analogously, $X_{\text{past}}$ contains as rows the vectors $x_{t,\text{past}}, t \in [T]$ and $\mathbf{X}_{\text{past}}$ multiple instantiations of $X_{\text{past}}$. With this notation, the SVAR (2) can be written concisely in the following matrix format:

$$X = X_{\text{past}} W + C \Leftrightarrow \mathbf{X} = \mathbf{X}_{\text{past}} W + \mathbf{C}. \tag{3}$$

The name root causes is motivated by the fact that the time-series data $X$ in (3) are uniquely determined by the input data $C$. Few root causes mean the input $C$ is sparse (Misiakos et al., 2023b). Intuitively, the nonzero values in $C$ represent unobserved events that propagate through space (according to $B_0$) and also through time $t$ to generate $X$ via (3). In practical applications the sparsity

---

[1] Vectors with negative indices are zero.

[2] Formally $W$ is not the adjacency matrix of the window graph, but contains all the necessary parameters to represent it. We will thus refer to the window graph with $W$.

[3] We will also always assume that (2) is stable for any time-series length $T$, i.e., that the data $X$ remain bounded. We provide a stability condition on $W$ in Appendix C.1.

in the root causes can only be satisfied approximately, thus we consider noise $N_c$ together with $C$ with zero mean and low standard deviation.

$$X = X_{\text{past}}W + C + N_c \Leftrightarrow \mathbf{X} = \mathbf{X}_{\text{past}}W + \mathbf{C} + \mathbf{N}_c. \tag{4}$$

In Fig. 1 we illustrate the data generation process (4). The root causes $C$ in the upper part are denoted in color, whereas white nodes correspond to approximately zero values (noise). The root causes percolate in time and space according to $W$ and generate the dense measured data $X$.

**Example.** In our stock market example, the root causes $c_t$ would represent significant events (big news) that trigger changes in the prices of the stocks at day $t$. Examples include unexpected quarterly results, administrative changes in the company, capital investment, lancing a new product, etc. It is intuitive that such events happen rarely and affect few stocks every day, and thus $C$ is sparse. Later, we confirm the few root cause assumption in experiments with real-world financial time series.

## 3  LEARNING THE SVAR

Here we show the identifiability of our setting, formulate a discrete optimization problem with the ground truth window graph as a global solution, and present our proposed method DAG-TFRC.

**Identifiability.** A natural question to ask in structure discovery is whether the graph is identifiable from data (Park, 2020). By this, we mean that there is a unique ground truth graph $W$ corresponding to the probability distribution of the data $\mathbf{X}$. Thus, the unknown graph can be uniquely determined from the data. Theorem 3.1 establishes the identifiability of $W$ in our setting.

**Theorem 3.1.** *Consider the time-series model (4). We assume that any root cause takes a uniform random value from $[-1, 1]$ with probability $p$[4] and is zero with probability $1 - p$. Then the adjacency matrices $B_0, B_1, ..., B_k \in \mathbb{R}^{d \times d}$ are identifiable from the time-series data $\mathbf{X}$.*

*Proof sketch.* The idea is to unroll $W$ over time into a DAG and rewrite (4) as a linear structural equation model (SEM) as explained in (Misiakos et al., 2024). Then, considering the (varying) locations of the non-zero entries in $\mathbf{C}$ modeled as Bernoulli random variables, the identifiability of the unrolled DAG is established from (Misiakos et al., 2023b, Theorem 3.1) which is a consequence of the non-Gaussian identifiability theorem by Shimizu et al. (2006). The window graph can then be identified by extracting $B_0, B_1, ..., B_k$ from the unrolled DAG. A full proof is in Appendix C.2. □

**Global minimizer.** Consider time-series data $\mathbf{X}$ generated with the SVAR (3), which is the noiseless version of (4). To solve for the window graph $W$ the motivation is that the root causes tensor $\mathbf{C}$ is assumed to be sparse and therefore the approximation $\widehat{W}$ should achieve as low number of non-zero root causes as possible. We thus propose approximating the window graph $W$ with the following discrete optimization problem:

$$\widehat{W} = \underset{W \in \mathbb{R}^{d(k+1) \times d}}{\arg\min} \|\mathbf{X} - \mathbf{X}_{\text{past}}W\|_0, \quad \text{s.t. } B_0 \text{ is acyclic}, \tag{5}$$

where $\| \cdot \|_0$ is the $L^0$ pseudonorm (number of non-zero entries). We can show that there is no other matrix than the ground truth $W$ that can achieve the lowest number of non-zero root causes. This result holds with high probability and in the case where the number of data is large enough and the root causes $\mathbf{C}$ are generated as in Theorem 3.1. Formally, the following Theorem 3.2 holds.

**Theorem 3.2.** *Consider time-series data $\mathbf{X}$ generated from (3). Then given a large (exponential in $dT$) amount of samples $N$, the matrix $W$ is with high probability the unique global minimizer of optimization problem (5).*

*Proof sketch.* As before, we unroll the window graph $W$ over time into a DAG and rewrite (4) as a linear SEM. Then the root causes satisfy the conditions of (Misiakos et al., 2023b, Theorem 3.2), and the ground truth unrolled DAG is the unique global minimizer. This also implies that $W$ is the unique global minimizer of (5). A complete proof is shown in Appendix C.3. □

---

[4]Low $p$ provides sparsity in $C$: few root causes.

**DAG-TFRC.** Based on the discrete optimization problem (5), we formulate our algorithm to learn the window graph $W$ from time-series data $X$ that are generated via (4). We refer to it as DAG-TFRC and it extends SparseRC in the same way as DYNOTEARS (Pamfil et al., 2020) extends NOTEARS (Zheng et al., 2018) to time series. Using continuous relaxation to handle the presence of noise and enforcing acyclicity with a continuous constraint $h(A)$ (e.g., NOTEARS (Zheng et al., 2018)) yields the following optimization problem:

$$\widehat{W} = \underset{W \in \mathbb{R}^{d(k+1) \times d}}{\arg \min} \|\mathbf{X} - \mathbf{X}_{\text{past}} W\|_1 + \lambda_1 \|W\|_1 + \lambda_2 \cdot h(B_0). \tag{6}$$

The first term in (6) promotes sparsity in the root causes and the other two terms promote the sparsity of the window graph $W$, and the acyclicity of $B_0$, respectively. Note that (6) is well suited for GPU operations with tensors, making it very efficient in practice. However, the implementation is non-trivial since it requires expressing $W$ with the parameters of a (PyTorch) linear layer with $(k+1)d$ inputs and $d$ outputs where the precomputed $\mathbf{X}_{\text{past}}$ is fed as input. The output of the linear layer is then subtracted from the data $\mathbf{X}$ and the computed objective in (6) is optimized with Adam optimizer (Kingma &Ba, 2014).

With $\widehat{W}$ computed via (6), we obtain an approximation $\widehat{\mathbf{C}}$ of the associated root causes via (7).

$$\widehat{\mathbf{C}} = \mathbf{X} - \mathbf{X}_{\text{past}} \widehat{W}. \tag{7}$$

$\widehat{\mathbf{C}}$ is an approximation since it is subject to the root cause noise $\mathbf{N}_c$ of the data generation equation (4). Thus, we also use thresholding to filter out the insignificant values in $\widehat{\mathbf{C}}$.

## 4 RELATED WORK

**Causal discovery from time series.** Causal discovery methods for time-series data aim to find the causal dependencies between nodes in time and recover either the window or summary graph which ignores the time delays (Gong et al., 2023). Many methods do not find dependencies that are truly causal (Assaad et al., 2022b) but rather discover spurious correlations (Assaad et al., 2022b).

Constraint-based methods utilize conditional independence tests to infer whether an edge exists between two nodes. Examples include Runge et al. (2019) that adapts the Peter-Clark (PC) algorithm (Spirtes et al., 2000) to time series as well as Entner &Hoyer (2010) do for the Fast Causal Inference (FCI) (Spirtes et al., 2000) to construct tsFCI. Other constraint-based methods are variations of PCMCI including (Runge, 2020; Gerhardus &Runge, 2020; Assaad et al., 2022a) or based on tsFCI (Malinsky &Spirtes, 2018).

Functional causal model-based methods consider the causal dependencies formulated according to a functional form. In this category, VAR-LiNGAM (Hyvärinen et al., 2010), models the data generation using an SVAR with non-Gaussian noise as input. This method extends LiNGAM (Shimizu et al., 2006) to time series and learns the window graph. TiMINO (Peters et al., 2013) and NBCB (Assaad et al., 2021) belong into the same category but only learn the summary graph. Closely related to functional causal models are methods that apply continuous optimization to learn the window graph. Such methods are extensions of NOTEARS (Zheng et al., 2018) to time series and include DYNOTEARS (Pamfil et al., 2020), NTS-NOTEARS (Sun et al., 2023) for non-linear data, and iDYNO (Gao et al., 2022) for interventional data.

Granger causality methods typically utilize the SVAR model to deduce the summary graph of pairwise Granger causal dependencies (Tank et al., 2021). (Pair-wise) Granger causality means that $i$ Granger-causes $j$ if the past values of the $i$th time series improve on the prediction of the present value of $j$th time series. Related works include neural Granger causality (Tank et al., 2021), eSRU (Khanna &Tan, 2019), GrID-Net (Wu et al., 2021), kernel (Marinazzo et al., 2008) and copula (Hu &Liang, 2014) Granger causalities and also convergent cross mapping (Sugihara et al., 2012). Another line of work employs neural networks for causal discovery from time series, like TCDF (Nauta et al., 2019), SCGL (Xu et al., 2019), neural graphical modelling (Bellot et al., 2022) and amortized learning (Löwe et al., 2022).

Extensions of the above methods consider irregular data (Cheng et al., 2023; 2024), subsampled data (Gong et al., 2015; Liu et al., 2023), or non-stationary time-series (Gao et al., 2024).

**Few root causes.** Our work falls into the category of continuous optimization methods but under the different data assumption of few root causes. Thus, our optimization objective differs from prior methods by minimizing the $L^1$ of the approximated root causes, whereas prior work employed the mean-square error loss. More comparison details are provided in Appendix B. The only method relying on the assumption of few root causes was SparseRC proposed by Misiakos et al. (2023b), designed for DAG learning from static data and successfully applied to gene data (Misiakos et al., 2023a). Here we advance over SparseRC in two aspects. The first is efficient, meaning fast and accurate, DAG learning from time series, particularly for larger DAGs and time lags. The second is applying the assumption of few root causes in financial data, broadening the range of its practical applications and for the first time providing a real interpretation on the approximated root causes. Misiakos et al. (2024) applied SparseRC for learning graphs from time series, by unrolling the window graph over time into a DAG, which requires learning $dT \times dT$ parameters. In this form SparseRC is not applicable to our experiments due to the resulting high complexity and times out, but we propose an alternative way of executing it to make the comparison feasible. However doing so cannot exactly mimic the assumed SVAR model. More details about this adaptation are in Appendix A.

**Root cause analysis.** The term root cause as used in our work is related to the use in prior root cause analysis (RCA) (Ikram et al., 2022). RCA is the process of determining the initial cause event that made some subcomponents of a system to fail. In RCA there is typically only one root cause; in our work there are multiple (but few) root causes in varying locations. RCA causes refer to defects or anomalies in the data, where in our case the root causes generate the entire measured data. Finally, the root cause in RCA is located at the closest (by some measure) ancestor of the failing nodes, where in our setting the root causes have a linear relation with the data. An application of our method to RCA may be a possible future direction.

## 5 EXPERIMENTS

We evaluate DAG-TFRC by comparing to prior state-of-the-art work on learning the window graph from time-series data. We consider synthetic, simulated and real data.

**Baselines.** We consider a variety of benchmarks ranging from linear data to non-linear that employ neural networks, considering only those that learn the full window graph. Concretely, we compare against VAR-LiNGAM, Directed VARLiNGAM (Hyvärinen et al., 2010), DYNOTEARS (Pamfil et al., 2020) using continuous optimization (like ours), NTS-NOTEARS (Sun et al., 2023) that assumes non-linear data generation, the constraint-based methods tsFCI (Entner &Hoyer, 2010) and PCMCI (Runge et al., 2019), as well as the deep learning method TCDF (Nauta et al., 2019).

We also compare against SparseRC, which is the only prior method explicitly imposing few root causes. SparseRC is designed for static data but has also been applied to time-series data in (Misiakos et al., 2024) by creating a static DAG by unrolling the time dimension. Doing so, SparseRC has to learn a DAG with $dT$ nodes, which, in our smallest experiment, already corresponds to $20 \cdot 1000$ nodes and times out. To give it a fair chance in our comparison we adapt its use to learn a smaller unrolled DAG corresponding to $k + 1$ time steps with $(k + 1)d$ nodes. The idea is to feed as input the data $\mathbf{X}_{\text{past}}$ and learn a matrix $\boldsymbol{A} \in \mathbb{R}^{(k+1)d \times (k+1)d}$, from which then the window graph $\boldsymbol{W}$ is extracted. With this formulation, we expect to compromise the performance of SparseRC but at least it becomes executable. More details are in Appendix A.

For the implementation, we use the publicly available repositories listed in Appendix D.9 with hyperparameters chosen with grid search listed in Appendix D.7 for synthetic and simulated data.

**Metrics.** We evaluate the unweighted approximation of $\boldsymbol{W}$ with the structural Hamming distance (SHD), i.e., the number of edge removals, insertions, and reverses needed to obtain the ground truth. In Appendix D.2 we include more metrics such as the area under ROC curve (AUROC), the F1 score and the normalized mean square error (NMSE), for the weighted approximation of $\boldsymbol{W}$. SID is computationally very expensive (times out) to run on DAGs with thousands of nodes and thus was not used. For the locations of the root causes in $\boldsymbol{C}$ we use again the SHD, computed as the total number of false or missed locations and the NMSE, shown in Appendix D.2, for the weighted approximation of $\boldsymbol{C}$. For every metric, we report the average and standard deviation (shown as

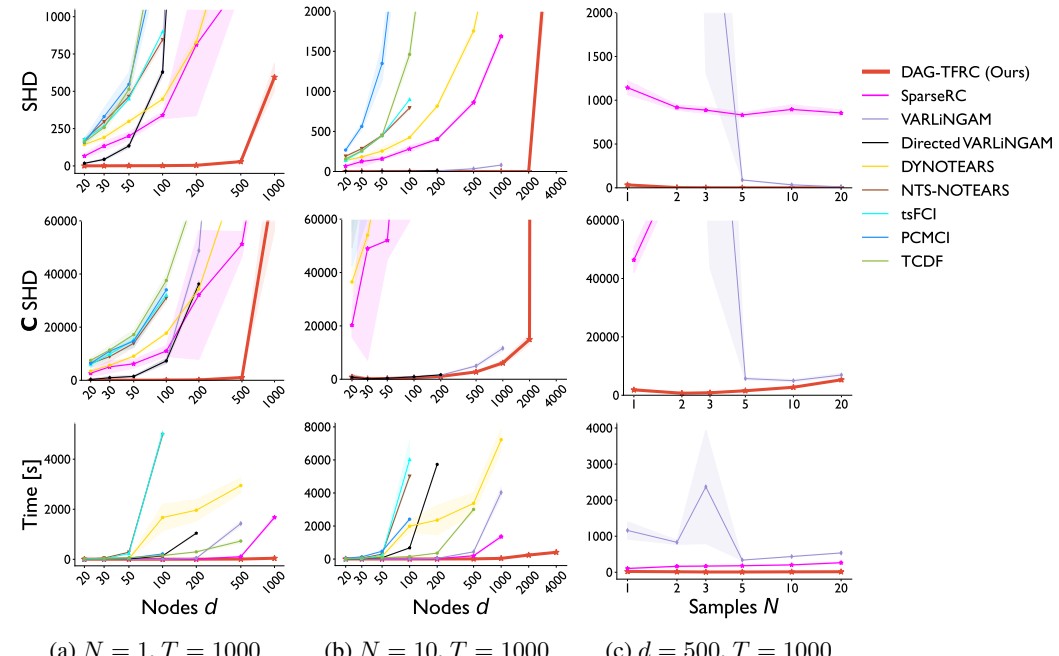

(a) $N = 1, T = 1000$    (b) $N = 10, T = 1000$    (c) $d = 500, T = 1000$

Figure 2: Performance on synthetic data. From top to bottom (lower is better): SHD, root causes SHD, and runtime. (a), (b) shows $N = 1$ and $N = 10$ samples of time-series with $T = 1000$ and varying number $d$ of nodes. (c) shows $d = 500$ nodes and varying samples $N$ of time-series of length $T = 1000$. Any non-reported point implies a time-out (execution time $> 10000s$).

shade in plots in Fig. 2) over five repetitions of the same experiment. In the real-world stock market dataset, the ground truth graph is unknown and thus the outcome can only be empirically evaluated.

## 5.1 SYNTHETIC EXPERIMENTS

**Data generation.** We use similar settings as (Pamfil et al., 2020) for the SVAR parameters and (Misiakos et al., 2023b) for the root causes, as explained next. First, we set the number of nodes $d$, the length of the time-series data $T$, the number of samples $N$ of the time series and the maximum lag $k$ of the SVAR model. For the window graph $\boldsymbol{W}$ we choose directed random Erdös-Renyi graphs $\boldsymbol{B}_0, \boldsymbol{B}_1, ..., \boldsymbol{B}_k$, where $\boldsymbol{B}_0$ is a DAG with average degree 5 and $\boldsymbol{B}_1, ..., \boldsymbol{B}_k$ have average degree 2. We consider a time lag of $k = 2$ by default and include a version with $k = 5$ in Appendix D.2. The edges of $\boldsymbol{W}$ are assigned uniform random weights from $[0.1, 0.5]$. Then, our time-series data are generated according to the SVAR in (4). With the upper bound $0.5$ in the weights of $\boldsymbol{W}$, the data $\boldsymbol{X}$ is likely bounded and we only discard it when its average value is higher than a large threshold (see Appendix D.1). The entries of $\boldsymbol{C}$ in (4) are non-zero with probability $p = 5\%$, in which case they are assigned a uniform weight from $[-1, -0.1] \cup [0.1, 1]$, otherwise they are $= 0$ with probability $1 - p$. We choose the threshold $0.1$ in (7) for all methods on evaluating the recovery of locations and time of the root causes. Finally, we consider $\boldsymbol{N}_c$ as zero-mean Gaussian with standard deviation $0.01$.

**Results.** Fig. 2 presents three variations of the synthetic experiment. Figs. 2a, 2b correspond to a fixed number of samples $N = 1, 10$ of time-series of length $T = 1000$. The number of nodes $d$ in each adjacency matrix $\boldsymbol{B}_0, \boldsymbol{B}_1, ..., \boldsymbol{B}_k$, ranges from 20 with 90 edges to 4000 with 36000 edges.

The SHD metric in Figs. 2a, 2b of PCMCI, TCDF, tsFCI and NTS-NOTEARS quickly deteriorates even for a small number of nodes. The same observation holds for $\boldsymbol{C}$ SHD which is anticipated as (7) requires a good approximation of $\boldsymbol{W}$. DYNOTEARS and SparseRC perform best among the baselines in Fig. 2a, but poorly in both 2a, 2b compared to DAG-TFRC. For DYNOTEARS this is expected as it promotes uniform and low-magnitude root cause values with the $L^2$-norm. But, it is surprising for SparseRC, and we believe the reason is computational errors because some

Table 1: SHD report for larger graphs. $T$ is set to 1000. Dashes represent timeouts $> 10000$s.

| Graph nodes $d$ | 1000 | | | | | 2000 | | | | | 4000 | | | | 8000 |
|---|---|---|---|---|---|---|---|---|---|---|---|---|---|---|---|
| Samples $N$ | 1 | 2 | 4 | 8 | 16 | 1 | 2 | 4 | 8 | 16 | 8 | 16 | 32 | 64 | 32 |
| DAG-TFRC (SHD) | 508 | 11 | **0** | **0** | **0** | 4890 | 929 | **2** | **0** | **0** | 4278 | 69 | 14 | **2** | - |
| VAR-LiNGAM (SHD) | - | - | - | 115 | 29 | - | - | - | - | - | - | - | - | - | - |

Table 2: Performance on the simulated financial dataset (Kleinberg, 2013).

| Method | SHD ($\downarrow$) | Time [s] |
|---|---|---|
| DAG-TFRC (Ours) | $11.36 \pm 7.46$ | $6.32 \pm 0.85$ |
| SparseRC | $\mathbf{9.92 \pm 8.22}$ | $9.74 \pm 1.21$ |
| VAR-LiNGAM | $19.25 \pm 10.64$ | $\mathbf{1.64 \pm 0.10}$ |
| Directed VARLiNGAM | $15.31 \pm 9.38$ | $4.85 \pm 0.31$ |
| TCDF | $19.06 \pm 10.18$ | $33.56 \pm 1.01$ |

time lags $\boldsymbol{B}_\tau, \boldsymbol{B}_{\tau+1}, ..., \boldsymbol{B}_k$, $\tau > 0$ are missing from the root cause equations, as explained in Appendix (A). In Fig. 2b, for $N = 10$ samples, the best competitors are VAR-LiNGAM and Directed VARLiNGAM and achieve a performance close to ours. But both are significantly slower than DAG-TFRC. Directed VARLiNGAM has a time-out on 500 nodes and VAR-LiNGAM on 2000 nodes. In particular, for 1000 nodes, VAR-LiNGAM is almost 1000 times slower than DAG-TFRC. In Fig. 2c, we set the number of nodes to $d = 500$ and vary the number of samples $N$ from 1 to 20 to see that VAR-LiNGAM requires many more samples than DAG-TFRC to perform reasonably well and is then always slower. Here, baselines that are not shown perform worse or have time-outs.

**Larger graphs.** We evaluate the performance of VAR-LiNGAM and DAG-TFRC for graphs with thousands of nodes and varying number of samples in Table 1. Those were the only methods without time-out and reasonable performance for 1000 nodes. First, we observe that VAR-LiNGAM times out for 2000 and 4000 nodes. For 1000 nodes we see that it requires more data time points to give a reasonable result. On the other hand, DAG-TFRC requires only a few samples on each case to achieve an excellent performance, except for 8000 nodes where it also times out. As one example we mention that DAG-TFRC can learn almost perfectly a window graph with $3 \times 4000$ (including the lags) nodes and $64 \times 1000$ data time points, in 2530s.

In Appendix D.2 we further include experiments on the sensitivity of the time lag $k$ showing that DAG-TFRC performance remains unaffected as long as it parametrizes a large enough time lag.

## 5.2 SIMULATED FINANCIAL PORTFOLIOS

We consider simulated financial time-series data from Kleinberg (2013). These represent daily stock returns generated with the Fama-French three-factor model (Fama, 1970) (volatility, size and value). The return $x_{i,t}$ of stock $i$ at time $t$ is computed as $x_{t,i} = \sum_j b_{ij} f_{t,i} + \epsilon_{t,i}$, where $f_{t,i}$ are the three factors, $b_{ij}$ are weights and $\epsilon_{t,i}$ are (correlated) idiosyncratic terms. Out of the available datasets, we use 16 that contain time lags up to 3 time steps. The data consists of daily returns of $d = 25$ stocks and the ground truth DAGs contain 22 edges on average. Each dataset contains a (multivariate) time series $\boldsymbol{X}$ with 4000 time steps which we split into 50 time-steps to create **X** of shape $80 \times 50 \times 25$.

In Table 2 we report the SHD and runtime. Note that here we cannot evaluate the root causes as there exist no ground truth ones. The hyperparameters of each method were chosen with grid search as explained in Appendix D.7. The top-performing methods are DAG-TFRC and SparseRC, indicating that few root causes is a valid assumption for these financial data. SparseRC performs slightly better than DAG-TFRC, which can be due to the very small scale: both time lag and number of nodes are small. However, SparseRC is still slower. The fastest method VAR-LiNGAM is worse in terms of performance. The other baselines didn't perform well and are left in the Appendix D.4.

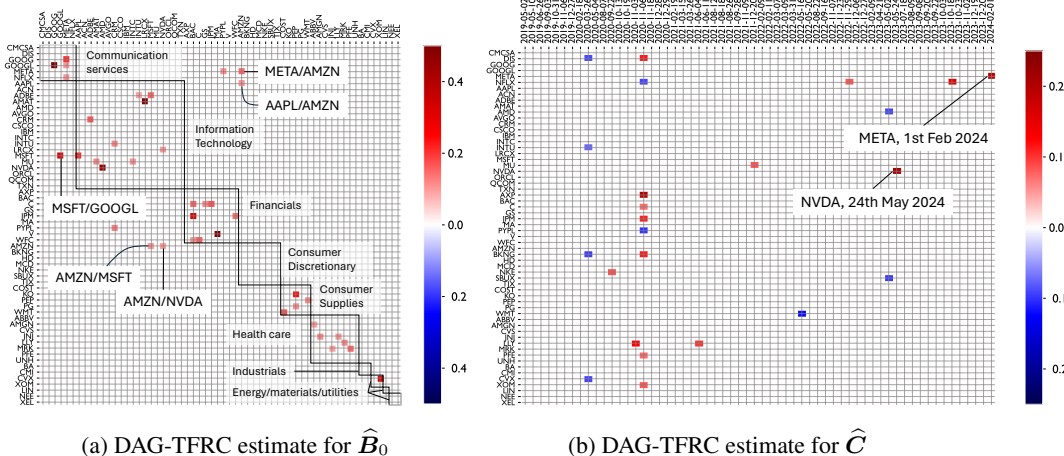

(a) DAG-TFRC estimate for $\widehat{\boldsymbol{B}}_0$    (b) DAG-TFRC estimate for $\widehat{\boldsymbol{C}}$

Figure 3: Real experiment on the S&P 500 stock market index. (a) Instantaneous relations $\widehat{\boldsymbol{B}}_0$ between the 45 highest weighted stocks within S&P 500, grouped by sectors (squares), and (b) the discovered root causes $\widehat{\boldsymbol{C}}$ for 60 days. In (a) the direction of influence is from row to column.

## 5.3  APPLICATION: S&P 500 STOCK DATA

**Dataset.** We consider stock values from the Standard and Poor's (S&P) 500 market index. A smaller version (S&P 100) of this index has been considered in (Pamfil et al., 2020). We gather data from the last 5 years: March 1st, 2019 to March 1st, 2024. We only consider the stocks that are present in the index during the entire time, which leaves $d = 410$ stocks as nodes. We collect the available daily close values for every stock which result in 1259 data time points for every stock and compute as data values the normalized log-returns (Pamfil et al., 2020). For the stock $i$ on day $t$ it is computed as $x_{t,i} = \log(y_{t+1,i}/y_{t,i})$, where $y_{t,i}$ is the closing value of stock $i$ at day $t$. We partition the time series into shorter time intervals of length 50 to obtain time-series data **X** of shape $25 \times 50 \times 410$. Using these data we learn the window graph that reveals the temporary relations between stocks and the underlying root causes that generate the data.

**Learning stock relations.** We execute all baselines with hyperparameters set according to the synthetic experiment. Fig. 3a shows the DAG-TFRC estimate for $\widehat{\boldsymbol{B}}_0$ representing the instantaneous relations between stocks. A figure with similar properties is discovered by SparseRC, but with the other baselines we did not manage to obtain reasonable result with our choice of hyperparameters or the ones from the published papers, see Appendix D.6. Below we analyze this result and argue that the few root cause assumption yields interpretable results for financial data.

For better visualization, we focus on the 45 highest weighted stocks in the S&P 500 index. In the execution of DAG-TFRC we set a maximum time lag $k = 2$, but it discovered that only $\boldsymbol{B}_0$ was significant. This confirms the known (Markovian) stock market principle: the previous day of the stock market contains all information from the past Fama (1970). Fig. 3a can be well interpreted. The edges of $\widehat{\boldsymbol{B}}_0$ roughly cluster the stocks w.r.t. their economic sectors. The few outliers are caused by the big IT companies being spread over three sectors as shown: (i) MSFT influences GOOGL (ii) META, AAPL both influence AMZN and (iii) AMZN influences MSFT and NVDA. Also, we note that the weights of $\widehat{\boldsymbol{B}}_0$ are positive which implies that stocks have a positive effect on each other: if one increases or decreases the others move along the same direction.

**Learning root causes.** From the window graph approximation $\widehat{\boldsymbol{W}}$, we can estimate the root causes with (7). In Fig. 3b we show the estimation for the same 45 stocks and 60 randomly chosen dates. As one may expect, they reflect significant changes in the stock price. To inspect further, we evaluated all computed root causes regarding their correspondence to stock price changes. We say that a root cause $c_{t,i}$ *aligns* with the change in data if $c_{t,i}\left[x_{t+1,i} - (1 + c_{t,i}/2)x_{t,i}\right] > 0$. For example, if a root cause equal to $+0.1$ aligns with the data change then $x_{t+1,i}$ is at least 1.05 times $x_{ti}$. Considering a threshold of 0.07 in $\widehat{\boldsymbol{C}}$ results in 4228 significant root causes which amounts to around 1% of the total time-series data points $NdT = 512500$. 99.6% of those aligned with the data changes. Thus,

whenever there is a significant root cause at day $t$, then the price of the stock at day $t + 1$[5] will increase if the root cause is positive (red) or decrease if the root cause is negative. Therefore, the root causes will reflect significant events affecting the stock values.

**News and dividends.** We conjecture that the root causes only correspond to significant changes that reflect unexpected events. For example META has a positive root cause $+0.18$ on $1^{st}$ Feb 2024 (Fig 3b). The same day META announced that it would pay dividends for the first time according to Reuters (2024). Similarly for the positive root cause $+0.21$ of NVDA on $24^{th}$ May 2023, the company announced jumps in its sales forecast as the demand for AI infrastructure increased, according to Reuters (2023). On the other hand, an event that is expected, but still affects the stock prices is the dividends, which are deducted on the ex-dividend date known well before. We conjectured that it is unlikely that a root cause reflects a dividend payment. In our dataset, we have a total of 3796 paid dividends, but only 35 of those coincided with a negative root cause, as conjectured.

**Limitations.** DAG-TFRC inherits limitations of structure learning based on SVAR. Using SVAR implies a linear model and stationarity, i.e., the window graph is the same for every time point and also across all time-series samples. The directed edges found are not necessarily true causal relations; establishing those would require further causal tools like interventions. We implicitly assume no undersampling: the measurement frequency is at least as high as the causal effects frequency. Undersampling may affect the stock market experiment where we used daily the measurements, but stock market effects happen within split seconds. In addition, we assume that there are no missing values in the data and the measurements on each node are taken with the same frequency. Also, while we can scale to thousands of nodes and better than prior work, very large graphs beyond that are still out of reach. Finally, we work with the underlying assumption of few root causes. In Appendix D.5 we include an experiment on the Dream3 challenge, a time series gene expression dataset. Our method is not the most appropriate, potentially because the linearity or the few root cause assumptions are violated.

## 6 CONCLUSION

Our main contribution to the body of work on causal inference from time-series data is the novel assumption of few root causes, which means that the data are generated via a small number of events in (node, time point) pairs. Assuming a standard SVAR model, we provided a practical algorithm that leverages this assumption to achieve both, higher accuracy and significantly faster execution on thousands of nodes than prior work as we illustrated in experiments. In particular, this included the accuracy in the discovery of the locations and time points of root causes. We motivated the few root cause assumption intuitively and with experiments on simulated and real financial data that yielded reasonable and interpretable results.

---

[5]The root cause effect happens on the next day as the data we consider are the log returns of stock prices.

## ETHICS

DAG-TFRC inherits the broader impact of other DAG learning methods from time series. From an ethical viewpoint, the methodology is generic and poses no specific potential risk.

## REPRODUCIBILITY

We acknowledge the importance of reproducibility and here we explain the actions that we took towards a more effortless reproduction of our results.

**Code.** We provide our code written in Python 3.9 as supplementary material and will make it available on github upon acceptance. In the README.md file, we explain the Python environment installation, how the code can be executed, and provide a Jupyter notebook demonstrating a synthetic experiment. More importantly, our code not only provides an implementation of our method but rather the whole experimental pipeline, showing how the data are generated and how the baselines are applied.

**Data.** The few root causes data generation can be executed using our code or reproduced according to the parameters explained in the experimental section of the main text and the details in Appendix D.1. For the simulated financial and the S&P 500 data we provide in Appendix D.10 the sources to download them.

**Methods.** We have explained in great detail in the main text the optimization problem solved by DAG-TFRC and the adapted version of SparseRC that we use for fair comparison, also explained in Appendix A. For the execution of all baselines, we use publicly available repositories listed in D.9 with hyperparameters set as shown in D.7. Competitor methods can also be executed using the provided code.

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

## A    APPLYING SPARSERC TO TIME-SERIES DATA

SparseRC (Misiakos et al., 2023b) is designed to learn a DAG from static data. Misiakos et al. (2024) applied SparseRC to learn graphs from time-series data by exploiting the structure of the unrolled DAG corresponding to the time series. For long time series, such a formulation creates a huge DAG to be learned - ranging from 20 thousand to 1 million nodes in our experiments. However, SparseRC can only be executed for $\approx 5000$ nodes at maximum to terminate in a reasonable time (Misiakos et al., 2023b). Thus it is impossible to be applied in our scenario in its prior form. For this reason, we propose an alternative way to apply SparseRC, which however, comes with a cost in approximation performance.

**SVAR as a Linear SEM.** To start with we show how an SVAR can be written as a linear structural equation model (SEM), which is the analogous model for generating linear static DAG data. We consider a time series $X$ generated with the SVAR in (3) (noiseless for simplicity). Consider the single-row vector $\boldsymbol{x} = (\boldsymbol{x}_0, \boldsymbol{x}_1, ..., \boldsymbol{x}_{T-1}) \in \mathbb{R}^{1 \times dT}$ consisting of the concatenation of the time-series vectors $\boldsymbol{x}_0, \boldsymbol{x}_1, ..., \boldsymbol{x}_{T-1}$ along the first dimension. Then (3) can also be encoded as:

$$\boldsymbol{x} = \boldsymbol{x}\boldsymbol{A} + \boldsymbol{c}, \tag{8}$$

where the root causes $\boldsymbol{c}$ here also have dimension $1 \times dT$. The matrix $\boldsymbol{A}$ is the adjacency matrix of a DAG with a special structure called the *unrolled DAG* (Kim &Anderson, 2012), which occurs by repeating the window graph corresponding to (2) for every time step $t \in [T]$:

$$\boldsymbol{A} = \begin{pmatrix} \boldsymbol{B}_0 & \boldsymbol{B}_1 & \dots & \boldsymbol{B}_k & \dots & \boldsymbol{0} \\ \boldsymbol{0} & \boldsymbol{B}_0 & \boldsymbol{B}_1 & & \ddots & \vdots \\ \vdots & \ddots & \ddots & \ddots & & \boldsymbol{B}_k \\ & & & \ddots & \ddots & \vdots \\ \boldsymbol{0} & & \dots & \boldsymbol{0} & \boldsymbol{B}_0 & \boldsymbol{B}_1 \\ \boldsymbol{0} & \boldsymbol{0} & & \dots & \boldsymbol{0} & \boldsymbol{B}_0 \end{pmatrix}. \tag{9}$$

This allows us to rewrite (3) as a linear structural equation model (SEM) (Shimizu et al., 2006):

$$\widetilde{\boldsymbol{X}} = \widetilde{\boldsymbol{X}}\boldsymbol{A} + \widetilde{\boldsymbol{C}}, \tag{10}$$

where $\widetilde{\boldsymbol{X}} \in \mathbb{R}^{N \times dT}$ consists of the $N$ time series as rows and $\widetilde{\boldsymbol{C}}$ is defined similarly for the root causes. Since $\boldsymbol{A}$ is a DAG, (10) represents a linear SEM.

**Original SparseRC.** We now explain how SparseRC can be applied to learn the window graph from time series according to Misiakos et al. (2024). SparseRC can be used to learn $\boldsymbol{A}$ from (many samples of) $\boldsymbol{x}$ stacked as a matrix $\widetilde{\boldsymbol{X}} \in \mathbb{R}^{N \times dT}$, generated from a linear SEM (10). Its optimization objective aims to minimize the number of approximated non-zero root causes $\widetilde{\boldsymbol{C}}$ in (10). This is expressed with the following discrete optimization problem

$$\widehat{\boldsymbol{A}} = \underset{\boldsymbol{A} \in \mathbb{R}^{dT \times dT}}{\arg\min} \left\| \widetilde{\boldsymbol{X}} - \widetilde{\boldsymbol{X}}\boldsymbol{A} \right\|_0, \quad \text{s.t. } \boldsymbol{A} \text{ is acyclic.} \tag{11}$$

The window graph $\widehat{\boldsymbol{W}}$ can be then extracted from the first row of the approximated $\widehat{\boldsymbol{A}}$. SparseRC in practice uses a continuous relaxation to solve optimization problem (11), but here we keep the discrete formulation for simplicity.

It can be seen that the DAG $\boldsymbol{A}$ consists of $dT$ nodes. In our smallest experiment this equals to $20 \times 1000 = 20000$ nodes, which is already out of reach for SparseRC. In contrast, DAG-TFRC requires to learn only $(k+1)\times$ DAGs with $d$ nodes each. Thus, we necessarily need to formulate SparseRC differently to be able to compare against it.

**Modified SparseRC.** The idea is to reduce the size of $\boldsymbol{A}$ by getting rid of the $\boldsymbol{0}'$s in (9). Specifically, instead of feeding SparseRC $\widetilde{\boldsymbol{X}}$ we feed as input $\mathbf{X}_{\text{past}}$. The resulting algorithm aims to find an $\widehat{\boldsymbol{A}}$ according to:

$$\widehat{\boldsymbol{A}} = \underset{\boldsymbol{A} \in \mathbb{R}^{(k+1)d \times (k+1)d}}{\arg\min} \left\| \mathbf{X}_{\text{past}} - \mathbf{X}_{\text{past}}\boldsymbol{A} \right\|_0, \quad \text{s.t. } \boldsymbol{A} \text{ is acyclic.} \tag{12}$$

To be compatible with the data-generating process, the following structure is assumed for $\boldsymbol{A}$:

$$\boldsymbol{A} = \begin{pmatrix} \boldsymbol{B}_0 & \boldsymbol{0} & \cdots & \boldsymbol{0} & \boldsymbol{0} \\ \boldsymbol{B}_1 & \boldsymbol{B}_0 & \ddots & & \boldsymbol{0} \\ \vdots & \boldsymbol{B}_1 & \ddots & \ddots & \vdots \\ \boldsymbol{B}_{k-1} & & \ddots & \boldsymbol{B}_0 & \boldsymbol{0} \\ \boldsymbol{B}_k & \boldsymbol{B}_{k-1} & \cdots & \boldsymbol{B}_1 & \boldsymbol{B}_0, \end{pmatrix} \tag{13}$$

The optimization objective (12) is different from $\|\mathbf{X} - \mathbf{X}_{\text{past}}\boldsymbol{W}\|_0$ used from DAG-TFRC and promotes a different convention in the data generating process. In particular by setting $\widetilde{\mathbf{C}} = \mathbf{X}_{\text{past}} - \mathbf{X}_{\text{past}}\boldsymbol{A}$ the root cause $\widetilde{\boldsymbol{c}}_{t-j}$ corresponding to the position $j$ of row $t$ of $\boldsymbol{x}_{t,\text{past}} = (\boldsymbol{x}_t, \boldsymbol{x}_{t-1}, ..., \boldsymbol{x}_{t-j}, ..., \boldsymbol{x}_{t-k})$ of a sample $i$ of $\mathbf{X}_{\text{past}}$ would be:

$$\widetilde{\boldsymbol{c}}_{t-j} = \boldsymbol{x}_{t-j} - \boldsymbol{x}_{t-j}\boldsymbol{B}_0 + \boldsymbol{x}_{t-j-1}\boldsymbol{B}_1 + ... + \boldsymbol{x}_{t-k}\boldsymbol{B}_{k-j} \neq \boldsymbol{c}_{t-j}. \tag{14}$$

This implies that the approximation of the root causes is not consistent with the data generation in (3), except when $j = 0$. Thus, only the first column of $\boldsymbol{A}$ promotes the correct equations and the rest undermine the performance of SparseRC. Resolving this discrepancy and keeping only the first column as trainable parameters is among the technical contributions of our paper.

## B   DAG-TFRC OPTIMIZATION AND COMPARISON WITH BASELINES

**DAG-TFRC.** Our implementation in PyTorch is outlined in Algorithm 1. It parametrizes the window graph matrix $\boldsymbol{W}$ using a single PyTorch linear layer and optimizes the objective function (6) with the Adam optimizer. The overall computational complexity of the algorithm is:

$$\mathcal{O}\left(M \cdot (NTd^2k + d^3)\right), \tag{15}$$

where $M$ is the total number of epochs (up to $10^4$).

The primary term in our objective, $\frac{1}{2NT}\|\mathbf{X} - L(\mathbf{X})\|_1$, represents a fundamental difference from prior work on causal discovery in time series. Methods such as VAR-based optimization approaches (Pamfil et al., 2020; Sun et al., 2023) typically rely on a mean-squared error loss supplemented by an $L^1$ penalty to promote sparsity in the DAG.

In contrast, both the main term and the regularizer in our objective are $L^1$ norms, promoting sparsity not only in the DAG but also in the root causes. This design aligns with the assumption of a few root causes in the data. Additionally, the $L^1$ norm has a constant gradient, ensuring consistent convergence speed. By comparison, $L^2$ minimization results in a diminishing gradient near the local optimum, potentially leading to longer convergence times.

### B.1   COMPARISON WITH BASELINES

**SparseRC.** As we explained in the main text, the method from Misiakos et al. (2023b) is infeasible to execute for long time series data. In its original form, SparseRC has complexity $\mathcal{O}\left(M \cdot (Nd^2T^2 + d^3T^3)\right)$, where $M$ is the total number of iterations. SparseRC learns a $dT \times dT$ unrolled DAG, which for our smaller scenario, results in a DAG with $d \times T = 20 \times 1000 = 20000$ nodes that goes beyond its computational reach (Misiakos et al., 2023b).

In Appendix A, we design a modified version of SparseRC that learns a $(k+1)d \times (k+1)d$ adjacency matrix, which ultimately leads to a complexity of $\mathcal{O}\left(M \cdot (NTd^2k^2 + d^3k^3)\right)$. This adaptation can be executed in most scenarios but comes at the cost of reduced model performance.

**VAR-LiNGAM.** First, the method fits a VAR model to the data:

$$\boldsymbol{x}_t = \widetilde{\boldsymbol{B}}_1\boldsymbol{x}_{t-1} + ... + \widetilde{\boldsymbol{B}}_k\boldsymbol{x}_{t-k} + \boldsymbol{n}_t, \tag{16}$$

and then performs Independent Component Analysis (ICA) to compute the self-dependencies matrix $\boldsymbol{B}_0$:

$$\boldsymbol{n}_t = (\boldsymbol{I} - \boldsymbol{B}_0)\boldsymbol{n}_t + \boldsymbol{c}_t. \tag{17}$$

**Algorithm 1** DAG-TFRC: DAG Learning from Time Series with Few Root Causes

---

**Input:** Time series data tensor $\mathbf{X} \in \mathbb{R}^{N \times T \times d}$, $\lambda_1, \lambda_2$ regularization parameters and threshold $\omega$.

**Output:** Weighted window graph $\widehat{W} = \begin{pmatrix} B_0 \\ \vdots \\ B_k \end{pmatrix}$ and root causes $\widehat{\mathbf{C}}$.

1: **Initialize:**
2: A single linear layer $L$(input: $d(k+1)$, output: $d$) in PyTorch that represents $\widehat{W}$.
3: Tensor $\mathbf{X}_{\text{past}} \in \mathbb{R}^{N \times T \times d(k+1)}$, where the $(n, t)$ entry is the vector $x_{t,\text{past}} = (x_t, x_{t-1}, ..., x_{t-k}) \in \mathbb{R}^{1 \times d(k+1)}$.

4: **Iterate:**
5: **for** each training epoch up to $M = 10^4$ **do**
6:    Compute the loss:
$$\frac{1}{2NT} \|\mathbf{X} - L(\mathbf{X})\|_1 + \lambda_1 \|W\|_1 + \lambda_2 h(B_0),$$
    where $h(B) = \text{tr}\left(e^{B \odot B}\right) - d$.
7:    Update the linear layer parameters $\widehat{W}$ with Adam optimizer.
8:    Stop early if the loss doesn't improve for 40 epochs.
9: **end for**

10: **Post-processing:**
11: Set the entries $w_{ij}$ of $W$ with $|w_{ij}| < \omega$ to zero.
12: Compute the unweighted version $U \in \{0, 1\}^{d(k+1) \times d}$ of $W$.
13: Compute the approximated root causes:
$$\widehat{\mathbf{C}} = \mathbf{X} - \mathbf{X}_{\text{past}} \widehat{W}.$$

14: **return** $\widehat{W}, \widehat{U}, \widehat{\mathbf{C}}$

---

The resulting matrices are calculated as

$$B_\tau = (I - B_0)\widetilde{B}_\tau.$$

The ICA step can be replaced with Direct LiNGAM (Shimizu et al., 2011), which guarantees convergence in a finite number of steps (under certain assumptions). This variation leads to the method Directed VARLiNGAM. However, both approaches have worse complexity compared to ours:

- For Direct LiNGAM: $\mathcal{O}\left(NTd^2k + NTd^3M^2 + d^4M^3\right)$, where $M$ is the number of iterations of Direct LiNGAM.
- For ICA LiNGAM: $\mathcal{O}\left(NTd^2k + NTd^3 + d^4\right)$, which lacks convergence guarantees.

In the large-DAG regime, these algorithms are inevitably slower than ours.

**DYNOTEARS.** Here, the mean-square error (MSE) is used, transforming the optimization into a quadratic problem:

$$\frac{1}{2NT} \|\mathbf{X} - \mathbf{X}W\|_2 + \lambda_w \|W\|_1 + \frac{\rho}{2} h(B_0)^2 + a h(B_0), \tag{18}$$

where the $L^2$ norm in the first term doesn't enforce sparsity on the root causes. As a result, this method experiences longer convergence times and produces a poor approximation of the ground truth window graph.

**TCDF.** This method fits convolutional neural networks (CNNs) to predict the time series at each node, based on the time-series values of other nodes in previous time steps. The approximation is optimized using the MSE loss. However, both the non-linearity of CNNs and the MSE loss do not align with our data generation process, which limits the method's effectiveness for our specific task.

**NTS-NOTEARS.** Similar to TCDF, this method also uses CNNs and MSE loss to approximate the window graph. In addition, the acyclicity regularizer from NOTEARS is applied. For similar

reasons, we anticipate low performance in our experiments with this method as well, due to the mismatch between the assumptions of the method and the characteristics of our data.

**tsFCI, PCMCI.** For the constraint-based baselines, there is no clear comparison in terms of optimization. These methods rely on statistical independence tests to infer causal dependencies between nodes at different time points. Empirically, however, these methods perform poorly, likely due to their inability to determine the causal direction for every edge they discover.

## C  Proofs

In this section we present the proofs for the technical results of the paper regarding the stability of the time series, the identifiability of the window graph from time-series data, and the proof that the ground truth window graph is the global minimizer in (5).

### C.1  SVAR stability

Whenever a measurement can be taken in a system, stability in the measured data holds by definition. For example, temperature measurements or stock price markets are never unbounded. To ensure that the same happens for synthetic data, one needs to guarantee the stability of the data generation process. A few prior works mention stability (Gong et al., 2015; Khanna &Tan, 2019; Bellot et al., 2022; Malinsky &Spirtes, 2018), and here we want to acknowledge its importance.

Equation (4) can be viewed as a discrete-time multi-input multi-output (MIMO) system (Skogestad &Postlethwaite, 2005), in which the input is the root causes $\boldsymbol{C}$ (together with noise) and the output is the time-series data $\boldsymbol{X}$. As the time-series length $T$ in (2) increases, the values of $\boldsymbol{X}$ can get arbitrarily large. We desire to find a range of weights for the matrices $\boldsymbol{B}_0, \boldsymbol{B}_1, ..., \boldsymbol{B}_k$ that guarantees that our time-series data are bounded. In particular, we require a condition for the bounded-input bounded-output (BIBO) stability of this system. This has been already considered by Lütkepohl (2005) (linear case, for non-linear refer to (Saikkonen, 2001)). The proposed condition requires the roots of the reverse characteristic polynomial to have a modulus less than 1. Here, we prove a practical and intuitive condition for stability as a derivation of the (Lütkepohl, 2005) result.

**Transitive closure.** To begin, we introduce the definition of the weighted transitive closure of the unrolled DAG (9).

$$\widetilde{\boldsymbol{X}} = \widetilde{\boldsymbol{X}}\boldsymbol{A} + \widetilde{\boldsymbol{C}} \Leftrightarrow \widetilde{\boldsymbol{X}} = \widetilde{\boldsymbol{C}}\left(\boldsymbol{I} - \boldsymbol{A}\right)^{-1} = \widetilde{\boldsymbol{C}}\left(\boldsymbol{I} + \overline{\boldsymbol{A}}\right), \tag{19}$$

On the right hand (19) $\overline{\boldsymbol{A}} = \boldsymbol{A} + ... + \boldsymbol{A}^{dT-1}$ is the weighted transitive closure (Seifert et al., 2023) of the unrolled DAG $\boldsymbol{A}$.

**Stability of model (4).** We will now prove Theorem C.1 that we are interested in. This provides a sufficient condition under which the model (4) is BIBO stable. BIBO stability here means that if the input root causes $\boldsymbol{C}$ and the noise $\boldsymbol{N}_c$ are bounded, then so are the output measurements $\boldsymbol{X}$.

**Theorem C.1.** *The model (4) is BIBO stable if for some (sub-multiplicative) matrix norm $\|\cdot\|$:*

$$\|\boldsymbol{W}\| < 1$$

*Proof.* If $\|\boldsymbol{W}\| = \lambda < 1$ then from the structure of $\boldsymbol{A}$ also $\|\boldsymbol{A}\| = \|\boldsymbol{W}\| = \lambda < 1$. Therefore:

$$\left\|\boldsymbol{I} + \overline{\boldsymbol{A}}\right\| = \left\|\boldsymbol{I} + \boldsymbol{A} + ... + \boldsymbol{A}^{dT-1}\right\| \leq \sum_{t=0}^{dT-1} \|\boldsymbol{A}\|^t \leq \sum_{t=0}^{dT-1} \lambda^t \leq \sum_{t=0}^{\infty} \lambda^t = \frac{1}{1-\lambda} = M$$

Thus

$$\lim_{T \to \infty} \|\boldsymbol{X}\| = \lim_{T \to \infty} \left\|\left(\boldsymbol{I} + \overline{\boldsymbol{A}}\right)\left(\boldsymbol{C} + \boldsymbol{N}_c\right)\right\|$$

$$\leq \lim_{T \to \infty} \left\|\boldsymbol{I} + \overline{\boldsymbol{A}}\right\| \|\boldsymbol{C} + \boldsymbol{N}_c\|$$

$$\leq M \|\boldsymbol{C} + \boldsymbol{N}_c\|$$

This implies that $\|\boldsymbol{X}\|$ is bounded for all $T$ and the model (4) is BIBO stable. $\qquad\square$

**Example.** Consider the induced $L^\infty-$norm as $\|\boldsymbol{A}\|_\infty = \max_j \sum_{i=1}^d |a_{ij}|$. The induced $L^\infty-$norm is sub-multiplicative and thus Theorem C.1 can be utilized. In fact it can be proved that any induced vector norm is sub-multiplicative (Theorem 5.6.2 in (Horn &Johnson, 2012)). Then, condition $\|\boldsymbol{W}\|_\infty < 1$ translates to all outcoming weights (rows of the window graph matrix) having the sum of absolute values less than 1.

For the sake of completeness, we provide a proof of the submultiplicativity property of the $L^\infty-$norm in Lemma C.2.

**Lemma C.2.** *The induced $L^\infty-$norm is submultiplicative.*

*Proof.* Consider any two square matrices $\boldsymbol{A}, \boldsymbol{B} \in \mathbb{R}^{d\times d}$. We need to show that $\|\boldsymbol{AB}\| \le \|\boldsymbol{A}\|\,\|\boldsymbol{B}\|$. Indeed,

$$\|\boldsymbol{AB}\| = \max_i \sum_{j=1}^d \left| \sum_{k=1}^d a_{ik} b_{kj} \right|$$

$$\le \max_i \sum_{j=1}^d \sum_{k=1}^d |a_{ik} b_{kj}|$$

$$= \max_i \sum_{k=1}^d \sum_{j=1}^d |a_{ik}|\,|b_{kj}|$$

$$= \max_i \sum_{k=1}^d |a_{ik}| \left( \sum_{j=1}^d |b_{kj}| \right)$$

$$\le \max_i \sum_{k=1}^d |a_{ik}| \left( \max_k \sum_{j=1}^d |b_{kj}| \right)$$

$$= \max_i \sum_{k=1}^d |a_{ik}|\,\|\boldsymbol{B}\|$$

$$\le \|\boldsymbol{A}\|\,\|\boldsymbol{B}\|$$

$\square$

**Example.** The $L^\infty-$norm is particularly interesting for our scenario as the condition of Theorem C.1 provides an intuitive interpretation for the weights. Consider our stock market example. Then the condition in C.1 means that for every stock that affects a set of other stocks, each with some factor $< 1$, the total sum should be less than 1. Of course, this is only a sufficient condition for the data to be bounded, but we believe that it is meaningful to consider that the influences between stocks are of this form in reality. To understand better why the condition in C.1 provides bounded data, we can think about it in the following way. When the $L^\infty-$norm is bounded, the total effect of a stock is divided into individual fractions that affect other stocks and doesn't get iteratively increased (which could be the case with sum $L^\infty-$norm $> 1$).

Bounding the sum of outcoming weights to 1 has also been considered in (Seifert et al., 2023; Misiakos et al., 2023b) in the scenario of pollution propagation in a river network.

## C.2 SVAR IDENTIFIABILITY

**Theorem C.3.** *Consider the time-series model (4). We assume that any root cause takes a uniform random value from $[-1, 1]$ with probability $p$ and is zero with probability $1 - p$. Then the adjacency matrices $\boldsymbol{B}_0, \boldsymbol{B}_1, ..., \boldsymbol{B}_k \in \mathbb{R}^{d\times d}$ are identifiable from the time-series data $\mathsf{X}$.*

*Proof.* As explained in Appendix A we can rewrite the SVAR (4) as a linear SEM:

$$\mathsf{X} = \mathsf{X}_{\text{past}} \boldsymbol{W} + \mathsf{C} + \mathsf{N}_c \Leftrightarrow \widetilde{X} = \widetilde{X} \boldsymbol{A} + \widetilde{C} + \widetilde{\boldsymbol{N}}_C. \tag{20}$$

The root causes follow a non-Gaussian distribution which implies that $\widetilde{C} + \widetilde{N}_C$ is also non-Gaussian. Moreover, based on the acyclicity assumption on $\boldsymbol{B}_0$, the unrolled matrix $\boldsymbol{A}$ represents a DAG and therefore (10) describes an SEM with non-Gaussian noise variables as in (Shimizu et al., 2006). The proof then follows from Theorem 3.1 in (Misiakos et al., 2023b) or similarly from the identifiability result in (Shimizu et al., 2006). Moreover, identifiability on $\boldsymbol{A}$ in turn implies identifiability for the parameters $\boldsymbol{B}_0, \boldsymbol{B}_1, ..., \boldsymbol{B}_k$ of the window graph $\boldsymbol{W}$, as desired. □

### C.3 GLOBAL MINIMIZER

**Theorem C.4.** *Consider time-series data* $\mathbf{X}$ *generated from (3). Then given a large (exponential in* $dT$*) amount of samples* $N$*, the matrix* $\boldsymbol{W}$ *is with high probability the unique global minimizer of optimization problem (5).*

*Proof.* As explained in Appendix A we can rewrite the SVAR (4) as a linear SEM: We will use again the equivalent of the SVAR as a linear SEM with the unrolled DAG.

$$\mathbf{X} = \mathbf{X}_{\text{past}}\boldsymbol{W} + \mathbf{C} + \mathbf{N}_c \Leftrightarrow \widetilde{\boldsymbol{X}} = \widetilde{\boldsymbol{X}}\boldsymbol{A} + \widetilde{C} + \widetilde{\boldsymbol{N}}_C. \tag{21}$$

Since, by construction, it is true that

$$\|\mathbf{C} + \mathbf{N}_c\|_0 = \left\|\widetilde{C} + \widetilde{\boldsymbol{N}}_C\right\|_0, \tag{22}$$

from the previous equation, we get that:

$$\|\mathbf{X} - \mathbf{X}_{\text{past}}\boldsymbol{W}\|_0 = \left\|\widetilde{\boldsymbol{X}} - \widetilde{\boldsymbol{X}}\boldsymbol{A}\right\|_0. \tag{23}$$

The root causes $\mathbf{C}$ (or $\widetilde{C}$) are generated entry-wise, each with probability $1 - p$ being 0 and with probability $p$ taking a uniform value in $[-1, -0.1] \cup [0.1, 1]$. Also, the linear SEM matrix $\boldsymbol{A}$ generating the data according to (10) is acyclic because of our assumption that $\boldsymbol{B}_0$ is acyclic and its construction, as shown in (9). Therefore, the conditions of Theorem 3.2 in (Misiakos et al., 2023b) are satisfied. This means that given a large amount of data that is exponential in the dimension of $\boldsymbol{A}$ ($= dT$) and also depends on the probability $p$, then it is guaranteed with high probability that any global minimizer of optimization problem (5) is equal to the ground truth $\boldsymbol{A}$. That also implies that the global minimizer is unique.

Denote:

$$\widehat{\boldsymbol{A}} = \underset{\boldsymbol{A} \in \mathbb{R}^{dT \times dT}}{\arg\min} \left\|\widetilde{\boldsymbol{X}} - \widetilde{\boldsymbol{X}}\boldsymbol{A}\right\|_0, \quad \widehat{\boldsymbol{W}} = \underset{\boldsymbol{W} \in \mathbb{R}^{d(k+1) \times d}}{\arg\min} \|\mathbf{X} - \mathbf{X}_{\text{past}}\boldsymbol{W}\|_0$$

Then according to (Misiakos et al., 2023b) $\widehat{\boldsymbol{A}} = \boldsymbol{A}$, the ground truth DAG, which has the form of an unrolled DAG corresponding to $\boldsymbol{W}$. We derive

$$\underset{\boldsymbol{A} \in \mathbb{R}^{dT \times dT}}{\min} \left\|\widetilde{\boldsymbol{X}} - \widetilde{\boldsymbol{X}}\boldsymbol{A}\right\|_0 = \left\|\widetilde{\boldsymbol{X}} - \widetilde{\boldsymbol{X}}\widehat{\boldsymbol{A}}\right\|_0 = \left\|\widetilde{\boldsymbol{X}} - \widetilde{\boldsymbol{X}}\boldsymbol{A}\right\|_0 = \|\mathbf{X} - \mathbf{X}_{\text{past}}\boldsymbol{W}\|_0$$

It follows that the ground truth window graph $\boldsymbol{W}$ is the global minimizer $\widehat{\boldsymbol{W}}$. Indeed for shake of contradiction, if there is $\boldsymbol{W}'$ with

$$\|\mathbf{X} - \mathbf{X}_{\text{past}}\boldsymbol{W}'\|_0 < \|\mathbf{X} - \mathbf{X}_{\text{past}}\boldsymbol{W}\|_0$$

Then this would allow us to construct an unrolled DAG $\boldsymbol{A}'$ with

$$\left\|\widetilde{\boldsymbol{X}} - \widetilde{\boldsymbol{X}}\boldsymbol{A}'\right\|_0 = \|\mathbf{X} - \mathbf{X}_{\text{past}}\boldsymbol{W}'\|_0 < \|\mathbf{X} - \mathbf{X}_{\text{past}}\boldsymbol{W}\|_0 = \left\|\widetilde{\boldsymbol{X}} - \widetilde{\boldsymbol{X}}\boldsymbol{A}\right\|_0$$

which is absurd. This concludes the desired result. □

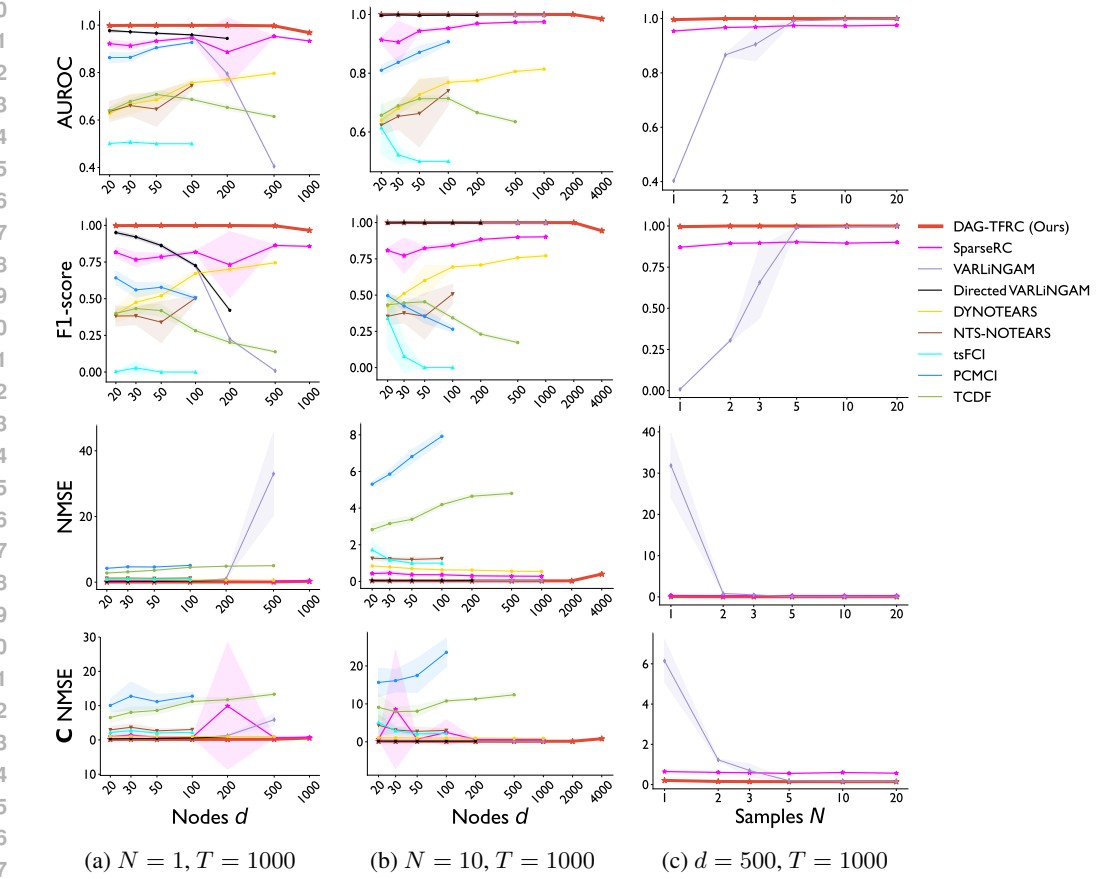

(a) $N = 1, T = 1000$    (b) $N = 10, T = 1000$    (c) $d = 500, T = 1000$

Figure 4: Performance on synthetic data: AUROC ($\uparrow$), F1-score ($\uparrow$) NMSE ($\downarrow$) and root causes NMSE ($\downarrow$). (a), (b) correspond to $N = 1$ and $N = 10$ samples of time-series with $T = 1000$ and varying number of nodes. (c) corresponds to $d = 500$ nodes and varying samples $N$ of time-series of length $T = 1000$. Any non-reported point implies execution time-out (over 10000s).

## D    EXPERIMENTS

### D.1    EXPERIMENTAL DETAILS

The stability of the time-series data $\boldsymbol{X}$ according to Theorem C.1 would require all $\boldsymbol{B}_0, \boldsymbol{B}_1, ..., \boldsymbol{B}_k$ to have an upper bound $w$ on their weights equal to $(5 + 2 + 2)w = 9w < 1$, or $w < 0.11$. Instead, to have a larger variety of weights we assign uniformly random weights from $[0.1, 0.5]$ to the edges. In practice, $\boldsymbol{X}$ most likely converges. Whenever we encounter unbounded data ($\boldsymbol{X}$ has an average value larger than $10^6 \cdot NdT$) we discard them and repeat the generation process.

### D.2    SYNTHETIC EXPERIMENTS

In Fig. 4 we provide the additional metrics AUROC (area under ROC curve), F1-score, the normalized mean square error (NMSE) and the NMSE on the root causes approximation. Formally, if $\widehat{\boldsymbol{W}}$ and $\widehat{\boldsymbol{C}}$ are the approximations of the ground truth window graph $\boldsymbol{W}$ and root causes $\boldsymbol{C}$ then:

$$\text{NMSE} = \frac{\left\|\widehat{\boldsymbol{W}} - \boldsymbol{W}\right\|_2}{\|\boldsymbol{W}\|_2}, \quad \boldsymbol{C}\,\text{NMSE} = \frac{\left\|\widehat{\boldsymbol{C}} - \boldsymbol{C}\right\|_2}{\|\boldsymbol{C}\|_2}. \tag{24}$$

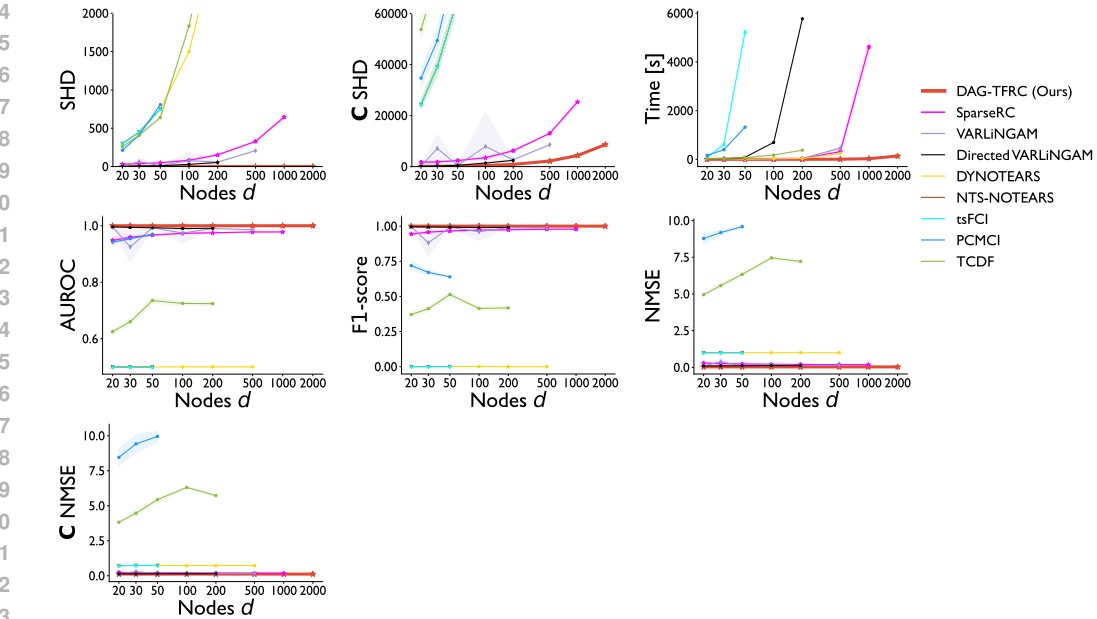

Figure 5: Performance on the experiment with more lags $k = 5$. The number of samples is set to $N = 10$ and each time series sample has length $T = 1000$. The plots show performance for varying number of nodes. The weight bounds are set to $[0.1, 0.2]$ for data convergence.

**Larger time lag.** In Fig. 5 we include an experiment with larger number of time lags $k = 5$. In this experiment, we have $N = 10$ samples of time series of length $T = 1000$ and varying number of nodes. The rest of the experimental settings are set the same as the main experiment. The only exception is the weight bounds for $W$ that here are set $[0.1, 0.2]$, because more lags imply that the weights need to be smaller to have bounded data according to the property of Theorem C.1. The results here are similar to Fig 2b. DAG-TFRC performs even better.

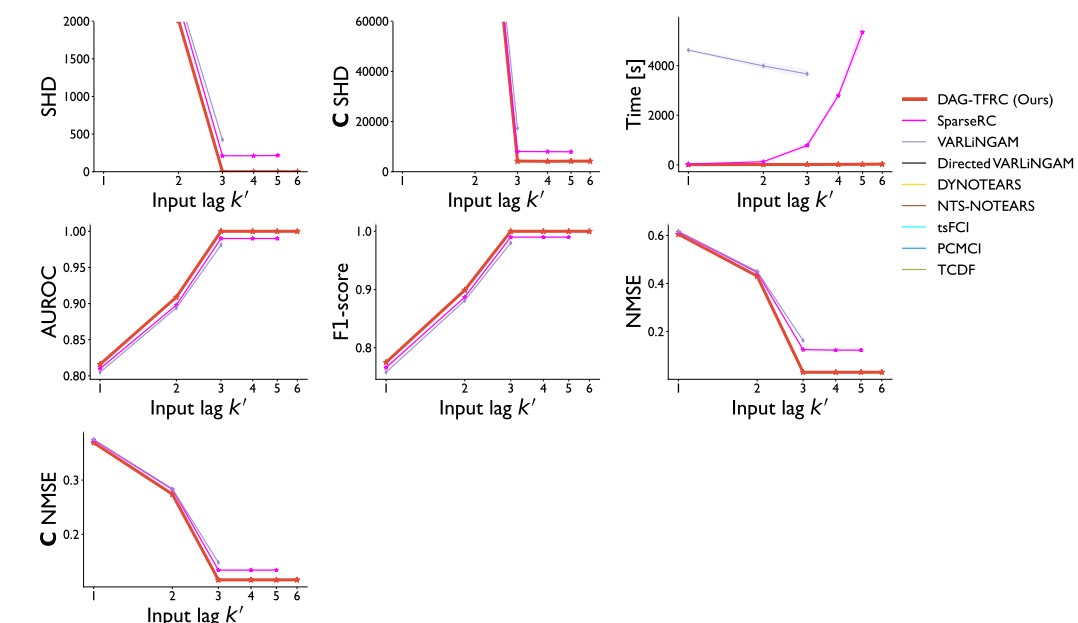

Figure 6: Evaluating the sensitivity of the time-lag $k$ in synthetic settings with original $k = 3$, $d = 1000$ nodes, $T = 1000$ and $N = 10$ samples. The algorithms have varying time-lag from 1 to 6.

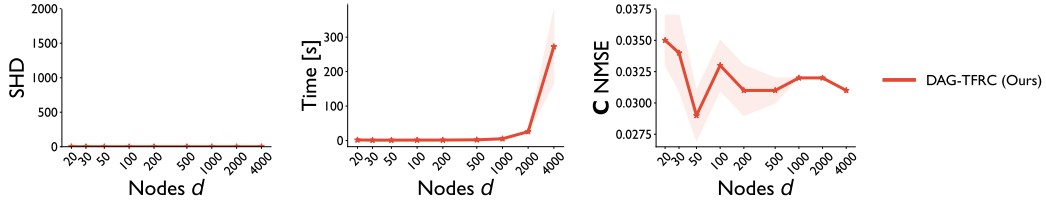

Figure 7: Evaluating the outcome of DAG-TFRC in the case where Gaussian noise $\mathcal{N}(0,1)$ is fed as input. We consider standard synthetic settings with original $k = 2$, $T = 1000$, $N = 1$ sample and varying number of nodes from 20 to 4000. The hyperparameters of DAG-TFRC are set to $\lambda_1 = 0.001 \cdot N \cdot T = 1, \lambda_2 = 1 \cdot N \cdot T = 1000$.

**Sensitivity of time lag.** We examine the sensitivity of the time lag parameter of the algorithms with the experiment in Fig. 6. In this experiment, we consider standard synthetic settings with $d = 1000$, $T = 1000$, and true time-lag $k = 3$.

When DAG-TFRC has time-lag $k' \geq k = 3$ as parameter, its approximation remains optimal. Therefore, given a large enough time-lag to DAG-TFRC, it can correctly detect the true underlying maximum time-lag $k$ of the system under consideration. We also saw this in our real experiment with stocks, in Fig. 3 where it didn't discover any time-lagged dependencies as expected (the stock market reacts almost instantaneously). As expected, DAG-TFRC performs poorly if the time-lag $k'$ given to the algorithm is smaller than 3.

SparseRC, performs well as long as $k' > k$, but still with worse approximation compared to DAG-TFRC. Moreover, it has higher execution runtime and times-out for $k' = 6$. VAR-LiNGAM performs reasonably when the algorithm is given the exact real time-lag $k$, but times-out when $k' > k = 3$.

Other baselines had a time-out or poor performance.

### D.3 GAUSSIAN ROOT CAUSES

We evaluate the behavior of DAG-TFRC in a special case scenario that doesn't obey the sparse root causes assumption. We consider that each entry in the input $\mathbf{X}$ is an independent noise variable $\sim \mathcal{N}(0, 1)$.

This scenario is equivalent to the data generation equation (4)

$$\mathbf{X} = \mathbf{X}_{\text{past}} W + \mathbf{C} + \mathbf{N}_c. \tag{25}$$

where $W = \mathbf{0}$ and $\mathbf{C} = \mathbf{0}$. Thus $\mathbf{X} = \mathbf{N}_c$. Our algorithm in this case will approximate the noisy root causes $\mathbf{C} + \mathbf{N}_c$ which are dense. Thus, in this case we re-weight the optimization objective (6) to give more emphasis in the terms $\lambda_1 \|W\|_1$ and $\lambda_2 \cdot h(B_0)$, in order to give $\widehat{W} = \mathbf{0}$ as output and less emphasis on the sparsity of root causes $\|\mathbf{X} - \mathbf{X}_{\text{past}} W\|_1$.

In Fig. 7 we show our results on this scenario. We choose hyperparameters $\lambda_1 = 0.001 \cdot N \cdot T = 1$ and $\lambda_2 = 1 \cdot N \cdot T = 1000$. With this approach, our algorithm manages to correctly find $\widehat{W} = \mathbf{0}$ and give a close approximation $\widehat{\mathbf{C}}$ on the ground truth root causes which are now equal to $\mathbf{C} + \mathbf{N}_c = \mathbf{N}_c$. Note, that in this trivial case it is crucial to account the noise in the root causes, as its contribution is very significant (large standard deviation of noise and empty root causes tensor).

### D.4 SIMULATED FINANCIAL DATASET

In Table 3 we show the results of the rest of the methods in the simulated financial dataset (Kleinberg, 2013). These methods perform worse for this setup, compared to methods utilizing the few root cause assumption.

Table 3: Performance on the simulated financial dataset Kleinberg (2013).

| Method | SHD ($\downarrow$) | Time [s] |
|---|---|---|
| DYNOTEARS | $33.92 \pm 9.09$ | $112.91 \pm 29.59$ |
| NTS-NOTEARS | $57.83 \pm 37.22$ | $16.40 \pm 14.45$ |
| tsFCI | $\mathbf{21.94 \pm 9.52}$ | $17.50 \pm 12.82$ |
| PCMCI | $361.69 \pm 67.80$ | $\mathbf{16.23 \pm 4.69}$ |

### D.5 DREAM3 CHALLENGE DATASET

Table 4: AUROC report on the Dream3 challenge dataset (Marbach et al., 2009; Prill et al., 2010). The methods are partitioned into non-linear and linear for a fair comparison. Best performances are marked with bold.

| | Model | E.coli-1 | E.coli-2 | Yeast-1 | Yeast-2 | Yeast-3 |
|---|---|---|---|---|---|---|
| | MLP | 0.644 | 0.568 | 0.585 | 0.506 | 0.528 |
| | LSTM | 0.629 | 0.609 | 0.579 | 0.519 | 0.555 |
| | TCDF | 0.614 | 0.647 | 0.581 | 0.556 | **0.557** |
| Non-linear | SRU | 0.657 | **0.666** | 0.617 | **0.575** | 0.55 |
| | eSRU | **0.66** | 0.629 | **0.627** | 0.557 | 0.55 |
| | PCMCI | 0.594 | 0.545 | 0.498 | 0.491 | 0.508 |
| | NTS-NOTEARS | 0.592 | 0.471 | 0.551 | 0.551 | 0.507 |
| | tsFCI | 0.5 | 0.5 | 0.5 | 0.5 | 0.5 |
| | **DAG-TFRC (Ours)** | 0.529 | 0.518 | **0.561** | 0.524 | 0.516 |
| | VARLiNGAM | 0.574 | 0.531 | 0.542 | 0.5 | **0.519** |
| Linear | Directed VARLiNGAM | 0.517 | 0.481 | 0.510 | 0.488 | 0.514 |
| | DYNOTEARS | **0.590** | **0.547** | 0.527 | **0.526** | 0.510 |

In Table 4 we report the AUROC performance of our method compared to baselines. There, Component-wise MLP and LSTM are from (Tank et al., 2021) and SRU and eSRU from (Khanna &Tan, 2019). while the rest of the methods are present in the main paper. The results of the first

5 rows are taken from (Khanna &Tan, 2019) and DYNOTEARS from (Gong et al., 2022). The methods are partitioned into non-linear and linear for a fair comparison.

Our method is competitive to other linear-model baselines but worse than those assuming a nonlinear model. Apparently, one of the two assumptions, either the few root causes assumption or linearity of the data generation does not hold in this dataset and our method might not be the most appropriate.

### D.6  S&P 500 REAL EXPERIMENT

In Figs. 8 and 9 we show the performance of SparseRC, VAR-LiNGAM, TCDF and PCMCI on the S&P 500 stock market index.

As also mentioned in the main text, SparseRC approximates a DAG similar to DAG-TFRC. This is due to the few root cause assumption that both methods use.

VAR-LiNGAM seems to identify significant edges for any random stock combination, thus producing a poor result. Also, the approximated root causes $\widehat{C}$ are less expressive than ours in the sense that out of the $4507$ discovered root causes only $33.7\%$ of them align with the data changes.

TCDF produces a very sparse DAG with not enough information.

Also, PCMCI outputs a zero graph for time-lag $0$ and a not well-structured graph for time-lag $1$. As a consequence, we don't see a meaningful pattern in the root causes.

DYNOTEARS had as output an empty graph and thus its performance is not reported. Regarding its hyperparameters, we minimized the weight threshold up to $0$ (all weights included as edges) and we tried both $\lambda_w = \lambda_a = 0.01$ and $k = 2$, which were the optimal from our synthetic experiments and $\lambda_w = \lambda_a = 0.1$ which is the reported best in the S&P 100 experiment in (Pamfil et al., 2020).

Directed VARLiNGAM, tsFCI and NTS-NOTEARS had time-out in this experiment.

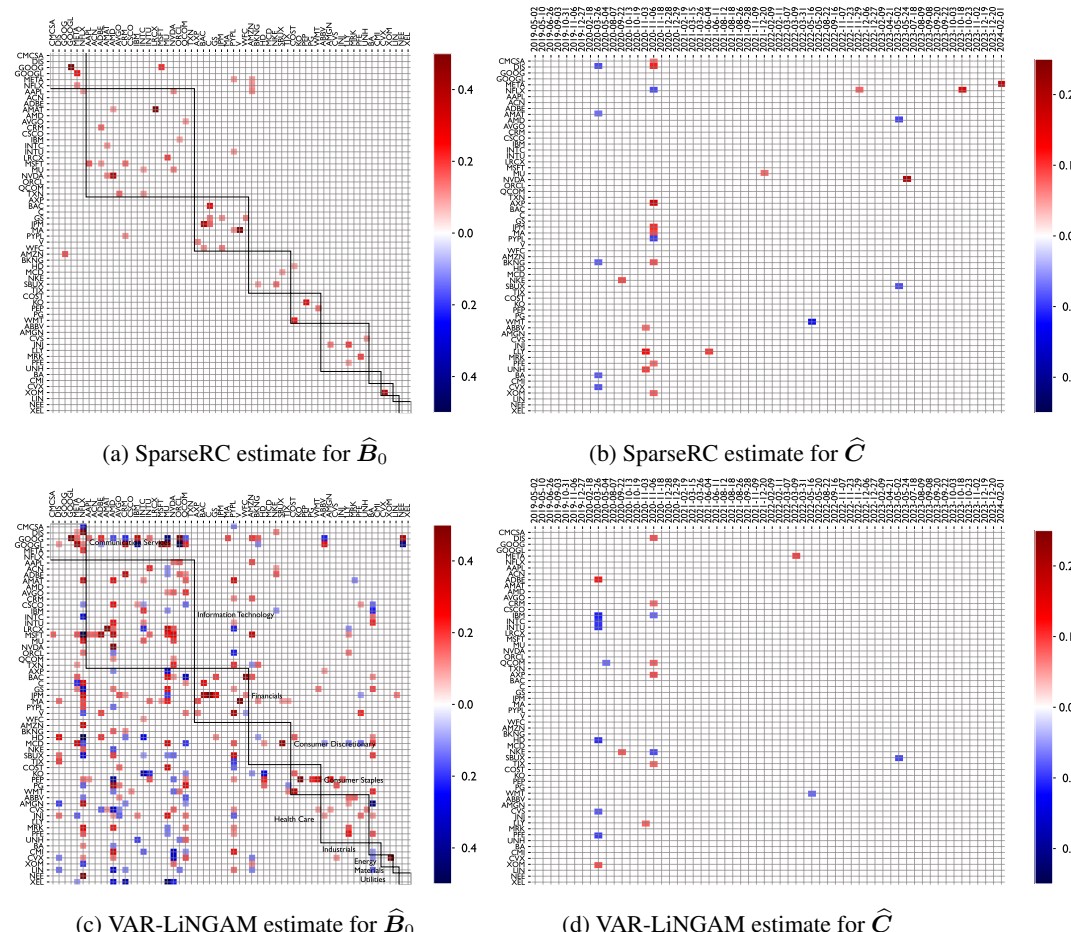

(a) SparseRC estimate for $\widehat{\boldsymbol{B}}_0$      (b) SparseRC estimate for $\widehat{\boldsymbol{C}}$

(c) VAR-LiNGAM estimate for $\widehat{\boldsymbol{B}}_0$      (d) VAR-LiNGAM estimate for $\widehat{\boldsymbol{C}}$

Figure 8: Evaluating baselines on the real experiment with S&P 500 stock market index. (a) Instantaneous relations between the $45$ highest weighted stocks within S&P 500 and (b) the discovered root causes for 60 dates.

## D.7  HYPERPARAMETER SEARCH

To find the most suitable hyperparameter selection for each method in our synthetic and simulated experiments we perform a grid search and choose the parameter combination that achieves the best SHD performance.

### D.7.1  SYNTHETIC EXPERIMENTS

For convenience we perform the grid search on small synthetic experimental settings ($N = 1$ sample, $T = 1000$ time steps, $d = 20$ nodes) where all methods have reasonable execution time. Note that for all methods we set their parameters regarding the number of lags correctly, to equal the ground truth lag (default $k = 2$). Also, the weight threshold is set to $0.09$ for all methods that compute the weighted adjacency matrix (true weights have magnitude ranging in $[0.1, 0.5]$). Any non-relevant hyperparameter that is not mentioned is set to its default value. The hyperparameter search gave the following optimal hyperparameters for each method:

**DAG-TFRC.** We set $\lambda_1 = 0.001$, $\lambda_2 = 1$ the coefficients for the $L^1$ and acyclicity regularizer, respectively. We let DAG-TFRC run for 10000 epochs, although usually it terminates earlier as we have an early stopping activated when for 40 consecutive epochs the loss didn't decrease.

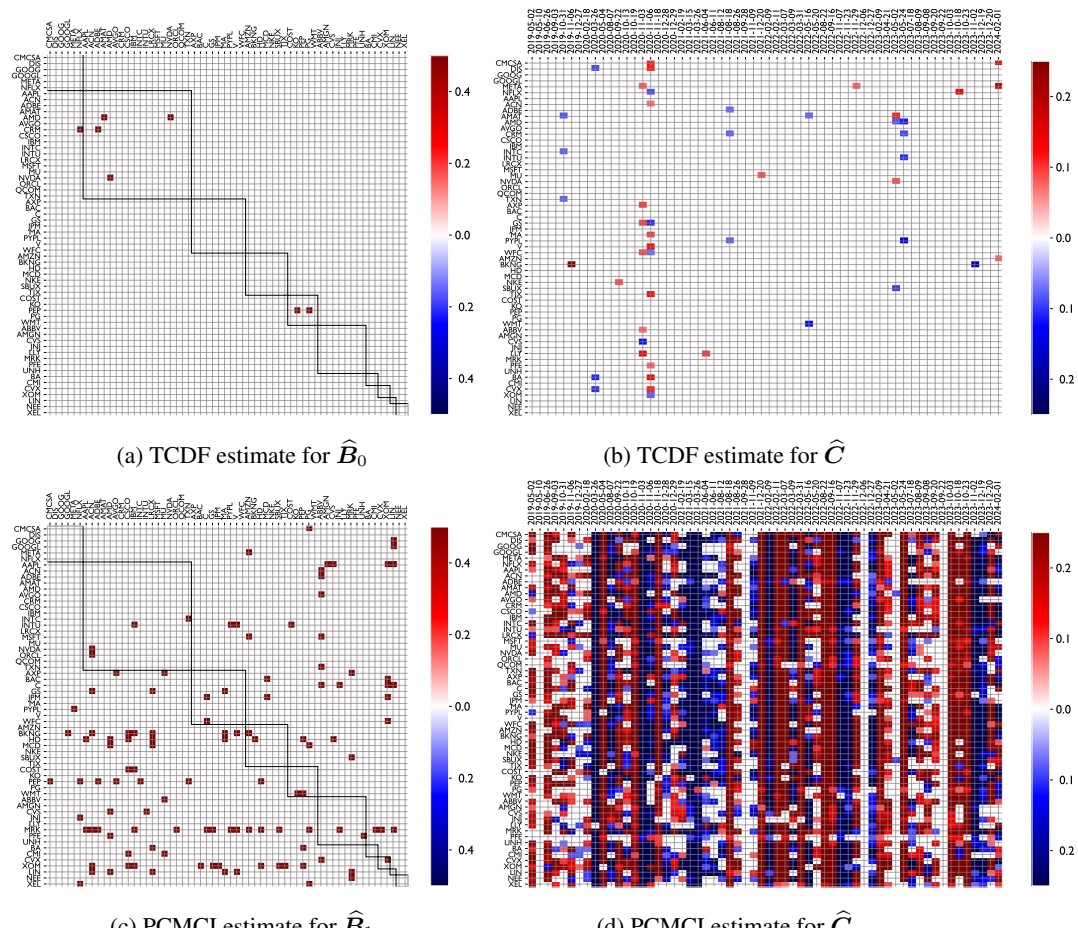

(a) TCDF estimate for $\widehat{\boldsymbol{B}}_0$      (b) TCDF estimate for $\widehat{\boldsymbol{C}}$

(c) PCMCI estimate for $\widehat{\boldsymbol{B}}_1$      (d) PCMCI estimate for $\widehat{\boldsymbol{C}}$

Figure 9: Evaluating PCMCI on the real experiment with S&P 500 stock market index. (a) Relations between the 45 highest weighted stocks within S&P 500 with time-lag 1 and (b) the discovered root causes for 60 dates.

**SparseRC.** We set $\lambda_1 = 0.001$, $\lambda_2 = 1$, $\lambda_3 = 0.001$ the coefficients for the $L^1$, acyclicity and block-Toeplitz regularizers, respectively. We similarly let SparseRC run for 10000 epochs, although usually it terminates earlier using early stopping as with DAG-TFRC.

**VAR-LiNGAM.** We may choose between ICA or Direct LiNGAM. In our experiments, we consider both cases (VAR-LiNGAM and Directed VARLiNGAM). The weight threshold is set to 0.09.

**DYNOTEARS.** The resulting values are $\lambda_w = \lambda_a = 0.01$.

**NTS-NOTEARS.** The resulting values are $\lambda_1 = 0.002$, $\lambda_2 = 0.01$. The $h_{tol}$ and the dimensions of the neural network were left to default.

**tsFCI.** Significance level is set to 0.1. Note that the output of tsFCI is a partial ancestral graph (PAG), which we therefore need to interpret as a DAG. For this scope we follow the rules of DYNOTEARS (Pamfil et al., 2020), meaning that whenever there is ambiguity in the directionality of the discovered edge we assume that tsFCI made the correct choice (this favors and over-states the performance of tsFCI). In particular, we translate the edge between nodes $i$ and $j$ in the following ways (i) if $i \rightarrow$ we keep it, (ii) if $i \leftrightarrow j$ in the PAG we discard it, (iii) either $i\circ \rightarrow j$ or $i \circ - \circ j$ we assume tsFCI made the correct choice, by looking at the ground truth graph.

**PCMCI.** The ParCorr conditional independence test was chosen. We do so because this test is suitable for linear additive noise models. Parameters are set as $pc_a = 0.1$, $a_{level} = 0.01$. The output can sometimes be ambiguous ($\circ - \circ$) because the algorithm can only find the graph up to the

Markov equivalence class, or there be conflicts $(x - x)$ in the conditional independence tests. In the former case, we assume that PCMCI made the correct choice and in the latter we disregard the edge.

**TCDF.** Here the kernel size and the dilation coefficient are set as the number of lags $+1$ ($k + 1 = 3$). The other parameters are $significance = 1$ and $epochs = 1000$.

### D.7.2 SIMULATED FINANCIAL DATA

Here we perform the grid search on the first available dataset of the simulated data (out of the 16 available) and choose the hyperparameters offering the best SHD performance. Here, we search for the most compatible weight threshold $\omega$ as the distribution of the ground truth weights is not known from the data generation. For all methods we set the number of maximum time lags at 3, which is the maximal ground truth lag. Any non-relevant hyperparameter that is not mentioned is set to its default value. The hyperparameter search gave the following optimal hyperparameters for each method:

**DAG-TFRC.** We set $\lambda_1 = 0.0001$, $\lambda_2 = 1$, $\omega = 0.5$. We let DAG-TFRC run for 10000 epochs at maximum.

**SparseRC.** We set $\lambda_1 = 0.001$, $\lambda_2 = 1$, $\lambda_3 = 0.1$, $\omega = 0.3$. We similarly let SparseRC run for 10000 epochs at maximum.

**VAR-LiNGAM.** The weight threshold is set to $\omega = 0.5$ for VAR-LiNGAM and $\omega = 0.6$ for Directed VARLiNGAM.

**DYNOTEARS.** The resulting values are $\lambda_w = 0.05$, $\lambda_a = 0.01$, $\omega = 0.3$.

**NTS-NOTEARS.** The resulting values are $\lambda_1 = 0.001$, $\lambda_2 = 1$, $\omega = 0.1$. The $h_{tol}$ and the dimensions of the neural network were left to default.

**tsFCI.** Significance level is set to 0.001 and $\omega = 0.1$ As previously we favor tsFCI in case of ambiguity, using the ground truth.

**PCMCI.** The ParCorr conditional independence test was chosen and parameters are set as $pc_a = 0.1$, $a_{level} = 0.01$, $\omega = 0.1$. In case of ambiguity, we assume PCMCI made the correct choice.

**TCDF.** The kernel size and the dilation coefficient are set as number of lags $+1$ ($k + 1 = 4$). The other parameters are $significance = 0.8$, $epochs = 1000$, $\omega = 0.2$.

### D.8 COMPUTE RESOURCES

Our experiments were run on a single laptop machine (Dell Alienware x17 R2) with 8 core CPU with 32GB RAM and an NVIDIA GeForce RTX 3080 GPU. The execution of the synthetic experiments for the 5 repetitions amounts to approximately 1 week of full run. Of course, initially there were some failed experiments, and after debugging the experiments were executed for only 1 repetition to determine where each method has a time-out. We thus chose the time-out to 10000 to try to make our experiments with as little cost as possible.

### D.9 CODE RESOURCES

For the implementation of the methods in our experiments we use the following publicly available repositories or websites. All github repositories are licensed under the Apache 2.0 or MIT license, except tigramite and TCDF which are under the GPL-3.0 license.

**SparseRC.** SparseRC code https://github.com/pmisiakos/SparseRC/. (MIT license)

**VAR-LiNGAM.** We use the official LiNGAM repo which we clone from github: https://github.com/cdt15/lingam. (MIT license)

**DYNOTEARS.** Code is available from the CausalNex library of QuantumBlack. The code is at https://github.com/mckinsey/causalnex/blob/develop/causalnex/structure/dynotears.py (Apache 2.0 license)

**NTS-NOTEARS.** We use the github code https://github.com/xiangyu-sun-789/NTS-NOTEARS provided by Sun et al. (2023). (Apache 2.0 license)

**tsFCI.** We use the R implementation from Doris Entner website which in turn utilizes the https://www.cmu.edu/dietrich/philosophy/tetrad/. Tetrad is licensed under the GNU General Public License v2.0. We also used the repository https://github.com/ckassaad/causal_discovery_for_time_series corresponding to the causal time series survey (Assaad et al., 2022b) (no license available).

**PCMCI.** We use the PCMCI implementation from (Runge et al., 2019) within the tigramite package. (GNU General Public License v3.0)

**TCDF.** We use the repository https://github.com/M-Nauta/TCDF from Nauta et al. (2019). (GNU General Public License v3.0)

**eSRU.** We use the repository https://github.com/iancovert/Neural-GC from Khanna &Tan (2019). (MIT License)

### D.10 DATA RESOURCES

**Simulated financial time series.** We take the data from http://www.skleinberg.org/data.html licensed under CC BY-NC 3.0

**S&P 500 stock returns.** The data are downloaded using *yahoofinancials* python library.

