# OpenReview forum: "Learning DAGs and Root Causes from Time-Series Data"
_ICLR.cc/2025/Conference — Submitted to ICLR 2025_

### Official Review · Reviewer_KUDz · 2024-11-01

**Soundness:** 2
**Presentation:** 2
**Contribution:** 2
**Rating:** 3
**Confidence:** 5

**Summary:**

This paper introduces DAG-TFRC, a method for learning directed acyclic graphs from time series data with few root causes, utilizing a structural vector autoregression model. The experiments were conducted on synthetic and real financial data to evaluate the effectiveness of this approach.

**Strengths:**

1. The experiments were carried out using both synthetic and real datasets.

2. Learning causal structures from time series data is an interesting and important problem.

**Weaknesses:**

1. **Technical Novelty and Contributions**
   - The proposed algorithm DAG-TFRC is just a scalable version of existing technique (SparseRC).
   - The technique contribution is limited or incremental. The authors claimed that their major contribution is "Our work expands the applicability of this assumption to the case of time series and, in addition, interprets the root causes in an experiment on real-world financial data", but the work of  Misiakos et al., 2024 (LEARNING SIGNALS AND GRAPHS FROM TIME-SERIES GRAPH DATA WITH FEW CAUSES) has already applied SparseRC for learning graphs from time series.  Then the only contribution seems to be "...interprets the root causes in an experiment on real-world financial data".

2.  **Overclaim in Title and Abstract**
    - The title and abstract suggest that the proposed method can learn both Directed Acyclic Graphs (DAGs) and root causes from time series data. If this is the case, the authors should compare their approach with not only existing DAG learning algorithms, but also the root cause analysis algorithms, especially causal structure learning based root cause analysis methods, in both the related work section and the experiments.
    - Consider changing the title, as it significantly overclaims the scope of this work.

3. **Writing and Presentation**
    - This paper is not well-written and not well-motivation, particularly the abstract and introduction. The introduction should be self-contained, but it fails to clarify the specific technical problem addressed, the motivation, why it is technically challenging, and what specific limitations of the existing approaches.
   - While the assumption of sparse root causes is intriguing, it lacks motivation in the context of time series.
   - The explanation regarding the maximum value of the time-lag 𝑘 is insufficient. More clarity is needed to understand its implications and significance.

4. **Experimental Section**
   - Figure quality is low: e.g., From Fig. 2, there are so many lines. It is hard to do comparison.
   - Different experimental settings are needed: e.g., only time-series of length T = 1000 is used in the experiments.
   - Limited results on the effect of the parameter k or how to set it in practice.
   - Current experimental results are unconvincing; only one real-world data (stock market) is used. The paper should include additional benchmark datasets (e.g., DREAM4 gene expression data) for a more comprehensive evaluation.

**Questions:**

1. The technical novelties and contributions, comparing to SparseRC (Misiakos et al. (2023b)) and Misiakos et al. (2024)?

2. In the related work, "...in our setting the root causes have a linear relation with the data", what do you mean by "root causes have a linear relation with the data"?

3. The authors claim that "SID is computationally very expensive (times out) to run on DAGs with thousands of nodes and thus
was not used", but why not trying different scales of graphs in the experiments? What the SID results would be on the small-scale graphs?

4. The authors claim that scalability is one of their contributions. Which specific aspects of the algorithm's design enhance its scalability compared to other methods?

---

> ### Author Response · Authors · 2024-11-21
> **Response to reviewer KUDz (part 1)**
>
> Dear reviewer KUDz,
>
> We have reported your review to the area chairs as inappropriate with a detailed explanation. Namely, your review is in large part **identical (copy paste)** or nearly identical to that we received from you for our previous submission to NeurIPS 2024. Moreover, you also included (exclusively) negative comments from the other previous reviewers, again some verbatim, which is inappropriate. All this despite the fact that we, of course, did not just resubmit, but substantially rewrote the entire paper for the new submission to address all previous reviewers concerns including yours.
>
> We will respond both to comments that have already been addressed in short and newly added comments.
>
> ## Already addressed comments
> **This paper is not well-written ... limitations of the existing approaches.**
> We had already rewrote the introduction according to your previous comments.
>
> - technical problem addressed -> Intro 1st par.: *"Our focus is structure discovery ... to reveal influence between temporal nodes.*. This has been updated now to *"Our work specifically focuses on learning these DAGs ..."*
> - technically challenging -> Intro 2nd par. : *"Learning a window graph with time lags is a challenging task ... NP-hard problem (Chickering et al., 2004)."*
> - limitations of existing approaches -> Intro 2nd par. : *"However, some methods, ... limited interpretation of the input variables of the SVAR, which we will refer to as root causes.*
>
> **While the assumption of sparse root causes is intriguing, it lacks motivation in the context of time series.**
> This is false. We have motivated the root causes assumption throughout the text with the stock market example.
>
> **The explanation regarding the maximum value of the time-lag 𝑘 is insufficient.** &
> **Limited results on the effect of the parameter k or how to set it in practice.**
> This is incorrect. From the previous submission we have already added an additional experiment. See Appendix D.2.
>
> **Current experimental results are unconvincing; only one real-world data (stock market) is used. ... comprehensive evaluation.**
> The stock market dataset is a challenging evaluation and from the few root causes assumption point of view it is meaningful and motivating.
> We added the Dream3 challenge in Appendix D.5. Apparently, our method is not the best but performs reasonably good. One of our two assumptions, few root causes or linearity is not true for this dataset. We have included this in the limitations paragraph.
>
> **The technical novelties and contributions, comparing to SparseRC (Misiakos et al. (2023b)) and Misiakos et al. (2024)?**
> See related work paragraph "few root causes", Appendix A and Appendix B.
>
> **The authors claim that "SID is computationally very expensive (times out) ... small-scale graphs?**
> Our goal is to show performance on the challenging large DAGs that naturally arise from time series.
>
> **The authors claim that scalability is one of their contributions. Which specific aspects ... compared to other methods?**
> Motivated by reviewer aws9 we have included a section (Appendix B) where we compare DAG-TFRC in terms of optimization and complexity with all baselines.

---

> ### Author Response · Authors · 2024-11-21
> **Response to reviewer KUDz (part 2)**
>
> ## New comments
>
> **The proposed algorithm DAG-TFRC is just a scalable version of existing technique (SparseRC).**
> This is not exactly correct. Indeed, SparseRC can be used for time-series, but DAG-TFRC is particularly designed for this case (as DYNOTEARS expanded NOTEARS for time series). We agree that DAG-TFRC is a more scalable version of both SparseRC and its adaptation in Appendix B.
>
> **The technique contribution is limited or incremental. ... an experiment on real-world financial data".**
> SparseRC has been used for time-series with very small time-length $T=10$ time-steps. In Appendix B we explain how cutting long time series in shorter intervals alters the data generating assumption. Thus our work is not incremental and we do provide significant changes in order to design an efficient DAG learning algorithm for time series.
>
> **The title and abstract suggest that the proposed method can learn both Directed Acyclic Graphs (DAGs) and root causes ... related work section and the experiments.**
> See related work, last paragraph. There we explain why root cause analysis (RCA) methods are incompatible for comparison.
> Applying our method to RCA is an interesting direction for future work.
>
> **Consider changing the title, as it significantly overclaims the scope of this work.**
> We don't agree, as we indeed find meaningful root causes in stock time series.
>
> **Figure quality is low: e.g., From Fig. 2, there are so many lines. It is hard to do comparison.**
> Please let us know if we could do something specific to improve readability. Our method is shown with the red line and we indicate which direction is better (low or higher) in the caption. Discarding a baseline from the plot is not an option as it would make our experiment less comprehensive.
>
> **Different experimental settings are needed: e.g., only time-series of length T = 1000 is used in the experiments.**
> Note that we have experiments with diffent number of samples ($10$ or $1$). Also the real experiment contains a different number of time steps ($T=50$).

---

> ### Comment · Reviewer_KUDz · 2024-11-26
>
> I would like to address the claims the authors made:
>
> 1. My review was updated based on your ICLR submission, not your NeurIPS submission, and is not "nearly identical" to my NeurIPS 2024 review.
>
> 2. If parts of my review remain similar, it reflects that the revisions have not sufficiently addressed the concerns and questions I previously raised, even if you believe otherwise.
>
> 3. I did not observe a major difference between the ICLR version and the NeurIPS version of the paper, which further supports my stance that the concerns remain unaddressed.
>
> 4. Regarding technical contributions and novelty, the authors claimed that their major contribution is "Our work expands the applicability of this assumption to the case of time series and, in addition, interprets the root causes in an experiment on real-world financial data",  and ""We refer to it as DAGTFRC and it extends SparseRC in the same way as DYNOTEARS (Pamfil et al., 2020) extends NOTEARS (Zheng et al., 2018) to time series", but the work of Misiakos et al., 2024 (LEARNING SIGNALS AND GRAPHS FROM TIME-SERIES GRAPH DATA WITH FEW CAUSES) has already applied SparseRC for learning graphs from time series. Thus, the authors' claim of novelty is fundamentally flawed, as Misiakos et al. (2024) had already applied SparseRC to time series, rendering their work unoriginal and misrepresented.
>
> I encourage the authors to focus on further refining the paper and sincerely addressing my feedback or questions, rather than making malicious and false accusations against the reviewer. For these reasons, I stand by my score.

---

### Official Review · Reviewer_BAnU · 2024-11-01

**Soundness:** 3
**Presentation:** 2
**Contribution:** 2
**Rating:** 6
**Confidence:** 3

**Summary:**

The authors propose DAG-TRFC, a method to learn the so-called window graph of a time series under the assumption it was generated by a structural vector autoregression (SVAR) model. From the learned window graph (which quantifies how much the value of a time series at $t$ is influenced by values at $t-k$), an approximation of a "root cause vector" can be derived. Loosely defined, a root cause is an event which significantly influences the observed time series. There may be up to $T$ (length of time series) root cause but the vector is typically sparse. Learning the window graph is framed as a discrete optimization problem and applied to synthetic and real world data thereby showing good performance a) in terms of smaller structural Hamming distance compared with other methods, and b) the recovery of interesting stock market events.

**Strengths:**

Without being an expert in root cause analysis for time series data, it seems the experiments are conducted in a sound and rigorous manner. Using SHD seems a reasonable metric and Figure 3 nicely illustrates the interpretability of the learned events.

**Weaknesses:**

The introduction is hard to parse. For a reading unfamiliar with SVAR models, it is extremely hard to understand the link between graphs and time series. To remedy this, the first paragraph could explicitly link the both concepts when mentioning the examples. The used graph terms should be directly linked to the time series domain. In addition, the first paragraph on Structural vector autoregression reads like a "related work" section rather than an introduction to the model itself. Despite the authors defining the term "root cause" several times in the manuscript, I still have a hard time grasping the meaning of a root cause in the time series context. I don't think it is a good term. Overall, the first half of section 2 is hard to follow (see questions below). Lastly, given that I can't develop an intuition for the root cause vector which is a central element of the work, I can't sufficiently judge the impact this method would have on the broader TS community.

**Questions:**

* Will the method always find a root cause? What would we discover on a time series containing only gaussian noise?
* My understanding is that $\mathbf{C}$ is of dimension $\[T, d\]$. How does this relate to the upper Figure 1?
* Eq. 1: Is it correct that $x_t$ is a function of itself? For VAR models that does not seem to be the case.
* Section 2, Time-series data should start with definitions over time series and then link to the graph notations. This interplay is hard to understand.
* When solving the optimization problem, how do you enforce the constraint that $B_0$ is acyclic?
* To learn root causes, you apply a threshold (page 8 last paragraph), how did you arrive at this and how general is it?
* In Figure 3, what is the direction of the influence in your legend (e.g., Meta/Amzn) would it be that meta influenced amzn or vice versa?

---

> ### Author Response · Authors · 2024-11-21
> **Response to reviewer BAnU (part 1)**
>
> Thank you for your comments regarding our writing and clarity. We note the following improvements on our text according to your feedback.
> - Introduction: we rewrote first paragraph and linked graph and time series clearer.
> - Section 2: we rewrote the first half to introduce better the notions of time series data, graphs and how those are linked with the SVAR model.
>
> The following responses refer to the updated version of the paper.
>
> **Despite the authors defining the term "root cause" several times in the manuscript, I still have a hard time grasping the meaning of a root cause in the time series context. I don't think it is a good term**
>
> Let us try to make root causes more understandable.
>
> Consider the SVAR Eq. (2) for $t=0$: $$\mathbf{x}_0 = \mathbf{x}_0 \mathbf{B}_0 + \mathbf{c}_0.$$
> Since, $\mathbf{B}_0$ is acyclic (a crucial commonly imposed assumption), there exist entries of $\mathbf{x}_0$ that correspond to nodes without parents (roots) and those are initialized with the value of $\mathbf{c}_0$ ($\mathbf{B}_0$ has a zero column on a root). The rest of the entries of $\mathbf{x}_0$ can be computed recursively using precomputed values of $\mathbf{x}_t$. More formally, $\mathbf{I} - \mathbf{B}_0$ is invertible, so the equation can be written as:
> $$\mathbf{x}_0 = \mathbf{c}_0 (\mathbf{I} - \mathbf{B}_0)$$
> implying that $\mathbf{x}_0$ is entirely determined by $\mathbf{c}_0$.
>
> In the general case, the recurrence in Eq. (2) can be solved similarly, which shows that all $\mathbf{x}_t$ are determined by all $\mathbf{c}_t$.  The data vectors at time points $t-1,...,t-k$ at are usually called causes of $\mathbf{x}_t$ but ultimately $\mathbf{x}_t$ is determined by the $\mathbf{c}_t$. That is why we refer to them as root causes, following prior work [1,2]. The stock market example is a good paradigm for the root causes, as the stock values are entirely determined from news that affect the stock market.

---

> ### Author Response · Authors · 2024-11-21
> **Response to reviewer BAnU (part 2)**
>
> ## Questions
> **Will the method always find a root cause? What would we discover on a time series containing only gaussian noise?**
>
> This is an interesting question for investigation. We have included this experiment in Appendix D.3.
>
> Your question is equivalent to assuming data generation with Eq. (2), but with all adjacency matrices $\mathbf{B}_{\tau}$ being zero and thus $\mathbf{x}_t = \mathbf{c}_t$, where the root causes $\mathbf{c}_t$ are generated as Gaussian noise. In essence this is what our approach finds. However, we found that since the sparsity assumption of the root causes doesn't hold, we needed to slightly adapt our algorithm's hyperparameters. Namely, we chose hyperparameters such that more weight of the optimization is given on the acyclicity term and the sparsity of the adjacency matrices and less weight on the sparsity of the root causes.
>
> **My understanding is that $C$ is of dimension $[T,d]$. How does this relate to the upper Figure 1?**
>
> Correct, for every possible node and time point combination there might exist a root cause, thus at most $dT$ root causes. In the upper graph of Fig. 1. every node represents a potential root cause. If it is white then it is approximately zero (no significant root cause), else it is positive or negative (significant root cause).
>
> **Eq. 1: Is it correct that $x_t$ is a function of itself? For VAR models that does not seem to be the case.**
>
> At first glance it looks like this, but no. The reason is that $\mathbf{B}_0$ is acyclic, so no entry of $\mathbf{x}_t$ depends on itself and the recurrence can be solved as explained above. Including $\mathbf{x}_t$ on the right hand side of Eq. (2) is exactly the difference between VARs and SVARs. Eq. (2) is solvable w.r.t to $\mathbf{x}_t$, if $\mathbf{B}_0$ is acyclic.
>
> **Section 2, Time-series data should start with definitions over time series and then link to the graph notations. This interplay is hard to understand.**
>
> Thanks for the feedback. You are right, it is better to explain this way. We have rewritten the explanation in Section 2, which now goes as follows. First, we introduce time series of multidimensional data. Then we impose the model assumed for generating the time series which introduces the graphs that describe how the values in a time step are obtained from prior time steps..
>
> **When solving the optimization problem, how do you enforce the constraint that is acyclic?**
>
> We enforce acyclicity using the regularizer $h(\mathbf{B}_0) - d = tr\left(e^{\mathbf{B}_0\odot \mathbf{B}_0}\right) -d$ which has been introduced by NOTEARS [1]. Intuitively, the $k$th term of the polynomial expansion of the exponential matrix penalizes cycles of length $k$. More details can be found in [1]. Our implementation and exact details of the use of the regularizer is shown in Appendix B, Algorithm 1.
>
> **To learn root causes, you apply a threshold (page 8 last paragraph), how did you arrive at this and how general is it?**
>
> The threshold is a parameter that we choose. We use the threshold $0.07$ to discard non-significant root causes. The significant root causes correspond to 1% of the total possible ones. Alternatively, we could have chosen to recover a larger percentage of root causes (e.g. 5% which was also the case in the synthetic experiments) by setting a threshold lower than $0.07$. This would additionally include smaller root causes, but still preserve the significant root causes that correspond to interesting events in the market.
>
> **In Figure 3, what is the direction of the influence in your legend (e.g., Meta/Amzn) would it be that meta influenced amzn or vice versa?**
>
> The direction is from row to column. For example (META, AMZN) means META affects AMZN. We added this in the caption of Fig. 3.
>
>
> [1] Xun Zheng, Bryon Aragam, Pradeep K Ravikumar, and Eric P Xing. DAGs with NO TEARS: Continuous Optimization for Structure Learning. Advances in Neural Information Processing Systems, 31, 2018.

---

> ### Comment · Reviewer_BAnU · 2024-11-25
> **Raising score**
>
> I thank the authors for their response and answering all my questions. All my issues are addressed and I will increase my score by 1 point. I still believe that manually selecting a threshold drastically hinders real-world applications and negatively impacts the quality of the work but I also see the merits of the proposed work.

---

### Official Review · Reviewer_aws9 · 2024-11-03

**Soundness:** 3
**Presentation:** 2
**Contribution:** 3
**Rating:** 6
**Confidence:** 4

**Summary:**

This paper proposes an L1-loss version of the vector autoregression (VAR) model to quantify the contribution of a non-autoregressive component, which the authors refer to as the "root cause."

While Granger causal learning using L2-loss combined with various regularizers has been extensively studied, the authors' motivation appears to be somewhat different.

The authors propose to use PyTorch (i.e., stochastic gradient) to solve the optimization problem but no details are provided.

**Strengths:**

- Tackles the relatively new problem of sparse causal learning.
- Presents a simple yet feasible formulation.

**Weaknesses:**

- Although the authors criticize existing methods as being ``generally inefficient for computing graphs with thousands of nodes,'' they do not provide detailed information on the optimization algorithm. It is mentioned that PyTorch was used, but Page 4 does not include specifics beyond this point.

- There is extensive literature on VAR-based causal learning with various regularization methods. While Section 4's coverage is adequate, it does not clearly distinguish the proposed method from existing approaches.

**Questions:**

Address the weakness points.

If the proposed method turns out to be really novel and effective from an optimization perspective in light of the existing works, I will raise the rating.

---

> ### Author Response · Authors · 2024-11-21
> **Response to reviewer aws9**
>
> Dear reviewer aws9, thank you for your insightful comments and suggestions.
> We have incorporated your feedback to improve both the presentation of our algorithm's implementation and its comparison with related work.
>
> Specifically, in the updated version, we have:
> - Expanded the few root causes paragraph to contrast better our method with prior work.
> - Included the pseudocode for the DAG-TFRC implementation in Appendix B for completeness.
>
> Below, we provide detailed responses to your comments, referring to the updated version of the paper.
>
> **Optimization Novelty**
> The novelty of our optimization Eq. (6) for the discovery of the window graph $\widehat{\mathbf{W}}$ from time-series data $\mathbf{X}$ lies in the use of the $L^1$ norm $\left\|\mathbf{X} - \mathbf{X}_{\text{past}}\widehat{\mathbf{W}}\right\|_1$ as the main term of our objective. This term corresponds to the root causes and the $L^1$ norm promotes sparsity, i.e., few root causes.
>
> In DYNOTEARS, a similar term was used for minimization but using the $L^2$ norm. Their assumption is that the root causes (as we call them) are independent zero mean noise variables with equal variances.
>
> We conjecture that the $L^1$ norm may accelerate convergence over the $L^2$ norm in the situation where the root causes are really sparse. In particular, the $L^2$ norm tends to promote uniformly distributed and therefore dense root causes, which will make it hard for the optimization to reach the ground truth in this case. Moreover, the $L^1$ norm has constant gradient and thus constant convergence speed, whereas the gradient of the $L^2$ norm diminishes near the local optimum, potentially leading to slower convergence.
>
> A rigorous formulation and proof of this conjecture, if possible, would be an interesting direction for future work.
>
>
> **Complexity and Optimization Comparison with Prior Work**
>
> **Our DAG-TFRC:** Our algorithm has a complexity of $\mathcal{O}\left(M \cdot (NT d^2 k + d^3)\right)$, where $M$ is the number of iterations of the optimization (epochs), $N$ is the number of samples $T$ is the length of the time series, $k$ is the maximum time lag, and $d$ is the number of nodes.
>
> **SparseRC:** In its original form of the published paper [2], it has complexity $\mathcal{O}\left(M \cdot (Nd^2T^2 + d^3T^3)\right)$ and thus for large $T$ (as used in our experiments) it times out. In our experiments, and for fairness, we used an obvious adaptation of SparseRC which we explain in Appendix A. Using it, we reduce SparseRC's complexity to $\mathcal{O}\left(M \cdot (NT d^2 k^2 + d^3k^3)\right)$ which is still slower than ours by a factor of $k^3$, but it computes an inaccurate approximation of the data-generating model (details in Appendix A).
>
> **VARLiNGAM & Direct VARLiNGAM:** The complexity of VARLiNGAM is the complexity of VAR $\mathcal{O}\left(NTd^2k\right)$ plus $\mathcal{O}\left(NTd^3 + d^4\right)$ if using ICA-LiNGAM or plus $\mathcal{O}\left(NTd^3M^2 + d^4M^3\right)$ if using the improved Direct LiNGAM ($M$ is the number of iterations). This explains why both are significantly slower than DAG-TFRC for large DAGs.
>
> **DYNOTEARS:** From an optimization perspective, it is similar to our approach, and thus shares the same runtime complexity. However, it uses the $L^2$ norm for the root causes, which is incompatible with the sparsity assumption. We conjecture that this leads to slower convergence (i.e., more epochs) and poorer approximations as confirmed in the experiments.
>
> **TCDF & NTS-NOTEARS:** Both are non-linear methods that employ convolutional neural networks to model dependencies between time series. However, they rely on the mean-square error ($L^2$) loss, which, we again conjecture, results in slow convergence and poor approximations.
>
> **PCMCI & tsFCI:** These are constraint-based methods that use conditional independence tests and cannot be directly compared in terms of optimization. Empirically, these methods perform poorly on our task.
>
> [1] Panagiotis Misiakos, Chris Wendler, and Markus Püschel. Learning DAGs from Data with Few Root Causes. Advances in Neural Information Processing Systems, 36, 2023.
>
> [2] Panagiotis Misiakos, Vedran Mihal, and Markus Püschel. Learning Signals and Graphs from Time-Series Graph Data with Few Causes. In ICASSP 2024-2024 IEEE International Conference on Acoustics, Speech and Signal Processing (ICASSP), pp. 9681–9685, 2024.

---

### Meta-Review · Area_Chair_rTTY · 2024-12-21

**Metareview:**

DAG-TFRC is introduced to learn DAGs from time series data with a few root causes. It is an adaptation of [Misiakos et al., NeurIPS 2023] for time series data; in particular, proofs of Theorems 3.1 and 3.2 are simply rearranging the data matrix so that  [Misiakos et al., NeurIPS 2023] can be directly applied. It is claimed that this adaptation renders more scalable implementation.

**Additional Comments On Reviewer Discussion:**

The authors complain that one the reviewers apparently also reviewed an earlier version of this work because the review looks similar. This alone does not invalidate the review. The AC themself read the submission and had a similar impression to KUDz that the novelty is limited and contribution is a little overstated.

---

### Decision · Program_Chairs · 2025-01-22

Reject